# ATTENTION AND COMPRESSION IS ALL YOU NEED FOR CONTROLLABLY EFFICIENT LANGUAGE MODELS

## ABSTRACT

The quadratic cost of attention in transformers motivated the development of efficient approaches: namely sparse and sliding window attention, convolutions and linear attention. Although these approaches result in impressive reductions in compute and memory, they often trade-off with quality, specifically in-context recall performance. Moreover, apriori fixing this quality-compute tradeoff in an architecture means being suboptimal from the get-go: some downstream applications require more memory for in-context recall, while others require lower latency and memory. Further, these approaches rely on heuristic choices that artificially restrict attention, or require handcrafted and complex recurrent state update rules, or they must be carefully composed with attention at specific layers to form a hybrid architecture that complicates the design process, especially at scale.

To address above issues, we propose **C**ompress & **A**ttend **T**ransformer (CAT), a conceptually simple architecture employing two simple ingredients only: dense attention and compression. CAT decodes chunks of tokens by attending to compressed chunks of the sequence so far. Compression results in decoding from a reduced sequence length that yields compute and memory savings, while choosing a particular chunk size trades-off quality for efficiency. Moreover, CAT can be trained with multiple chunk sizes at once, unlocking control of quality-compute trade-offs directly at test-time without any retraining, all in a single adaptive architecture.

In exhaustive evaluations on language modeling, in-context recall, and long-context understanding, a **single** adaptive CAT model outperforms many existing efficient models including hybrid architectures across varying inference budgets. Further, a single CAT model matches dense transformer in language modeling while being $1.4 - 3\times$ faster and requiring $2 - 9\times$ lower total memory usage.

## 1 INTRODUCTION

Transformers (Vaswani et al., 2017) are the default architectures for large language models (LLMs), and rely on powerful self-attention mechanism (Bahdanau et al., 2014). However, the compute required for decoding with dense self-attention grows quadratically with the sequence length, with memory costs growing linearly, making transformers expensive to deploy in a world, where inference cost dominate in the long run[1] (Timbers, 2023; Jassy, 2023; Bridi, 2025; GMI Cloud, 2025).

Given the cost of attention in transformers there has been interest in making them efficient. While approaches like sparse attention (Child et al., 2019; Zaheer et al., 2020) *heuristically* restrict the tokens being attended to, others like linear attention (Katharopoulos et al., 2020; Arora et al., 2024a; Dao & Gu, 2024; Yang et al., 2025b) use fixed-size recurrent states to enable constant compute and memory costs. However, restricting tokens apriori or using fixed-size recurrent states hurts in-context recall performance (Arora et al., 2024a; Jelassi et al., 2024; Wen et al., 2024). Learning to recursively and sequentially compress the sequence can avoid fixed-memory bottlenecks and heuristic restrictions (Rae et al., 2020; Chevalier et al., 2023), but sequential computations make the training slow and learning objective difficult to optimize (Geiping et al., 2025).

---

[1]in fact, the cost of training can break even with inference in *just* a few weeks (Timbers, 2023)

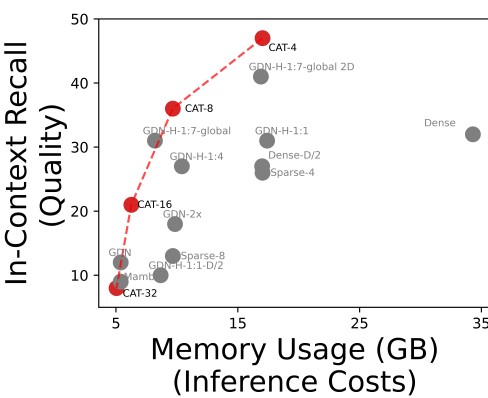

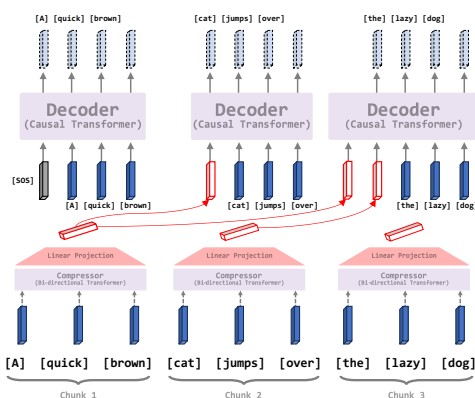

(a) **CAT unlocks test-time control of quality-inference cost trade-offs**, where a single adaptive CAT model (**red** dots) achieves a pareto frontier in quality-inference costs curve, outperforming nearly every popular architecture: **across 12 models** with different model configurations, parameter counts (∼300M to ∼820M) with **varying inference costs** on real-world in-context recall tasks. Further, a single CAT model outperforms most models in quality when latency budget matched (appendix Fig. 5).

(b) **The Compress and Attend Transformer (CAT) architecture.** CAT chunks up a sequence of length $N$ into $N/C$ chunks of $C$ tokens (illustrated for $C = 3$). Each chunk is parallelly compressed into a chunk representation. CAT then decodes each chunk by attending to past chunk representations. Compression results in a reduced sequence length enabling compute and memory savings during decoding. Chunk size in CAT acts as **knob**, offering test-time control of quality-efficiency trade-offs, where higher chunk sizes result in increased efficiency.

Figure 1

Moreover, not all downstream tasks have the same compute and memory requirements. For example, writing emails does not require strong in-context recall performance and linear attention may be a suitable choice but code autocompletion demands accurate recall of function names from the entire code repository in the context, requiring more memory and compute where dense attention may be preferred. The existing approaches for efficiency *fix* the compute and memory usage before training with choices like attention masks, window size or recurrent state size meaning if at test time a problem demands a higher budget for better performance, a whole new model needs to be trained. Training models with different tradeoffs is one way to tackle this problem but repeating this for every downstream task can become quickly prohibitive. Even if such models were available, learning to route between them based on the context requires holding all these models in memory.

To address the issues raised above, we propose a conceptually simple architecture: **C**ompress & **A**ttend **T**ransformer (CAT) that employs two simple well-known ingredients, namely dense attention and compression. CAT compresses chunks of tokens in parallel into a shorter sequence using a *compressor*, which a *decoder* then attends to while autoregressively modeling the tokens in the latest chunk; both compressor and decoder are simple dense transformers themselves (see Figure 1b). With the compression and decoding being parallel over tokens during training, there is no recurrence along the sequence dimension, which enables end-to-end **scalable training**. Decoding from the reduced sequence length due to compression enables **compute and total memory savings**. This reduction allows the use of more parameters in CAT, thereby improving model quality without increasing inference costs[2]. Choosing a particular chunk size in CAT trades-off quality for compute and memory (see Figure 1a). At the same time, the **memory grows *gracefully***, linearly with sequence length but at a significantly slower rate, to enable in-context recall performance at long sequence lengths. Importantly, training CATs across multiple chunk sizes at once **unlocks control of quality-compute trade-offs directly at test-time** without any retraining, all in a single adaptive architecture. Further, CAT at its **core** is a ***meta sequence mixer***, where the *compressor* and the *decoder* could make use of any existing sequence mixers (e.g. linear attention): meaning rather than being a *strict competitor*,

---

[2]similar in vein to flagship models (Yang et al., 2025a; Agarwal et al., 2025) that are all parameterized as Mixture-of-Experts (MoEs) (Shazeer et al., 2017), where additional model parameters does not imply increased inference costs, and brings improvements in quality.

**CAT is complementary to existing efficient sequence mixers.** Finally, CAT can be instantiated as a layer where it can be used as a drop-in replacement layer in any architecture (see Section 4)

To summarize, this paper:

- Introduces the CAT architecture to efficiently model sequences by decoding each chunk of tokens given parallelly compressed representations of the past chunks. Adjusting a single knob (chunk size) at test-time controls quality-efficiency trade-offs, allowing a single CAT model to *interpolate* between the dense transformer and efficient alternatives without any retraining.

- Provides a parallel and scalable implementation for training CATs (we scale from 90M to 1B parameters) and an efficient pure PyTorch implementation for generation that does not require any custom CUDA or Triton kernels, unlike most efficient baselines. We further provide an implementation for CAT as a drop-in replacement layer in any architecture.

- Demonstrates that a **single adaptive CAT model**
  - outperforms many popular efficient baselines including hybrid architectures on language modeling, common-sense reasoning, long-context understanding, in-context recall, and needle-in-haystack tasks, when inference costs matched, across varying inference budgets.
  - matches or outperforms the dense transformer on language modeling at multiple model scales while being $1.4 - 3\times$ faster and using a $2 - 9\times$ smaller total memory footprint.
  - surpasses, interestingly even the dense transformer on real-world in-context recall tasks using the least efficient setting (CAT-4) while still being atleast $1.5\times$ faster and $2\times$ memory efficient, akin to MoEs (Shazeer et al., 2017).

## 2 COMPRESS AND ATTEND TRANSFORMERS (CATS)

This section first describes components of the CAT architecture and how it's trained for test-time trade-offs between quality and compute. Second, it discusses CAT's practical implementation and the resulting compute and memory savings.

**Compression and decoding.** CAT uses a *compressor* $f_\theta$ and a *decoder* $g_\theta$, both instantiated as dense transformers. The *compressor* is a bidirectional transformer $f_\theta$ that has hidden size $D_f$, followed by a linear projection to $D_g$, and the *decoder* is a causal transformer $g_\theta$ having hidden size $D_g$, matching the linear projection from the *compressor*.

Given a sequence of $N$ tokens $\mathbf{x} = \{x_i\}_{i \leq N}$, we split the sequence into chunks of tokens, each of size $C$ represented by $\{\mathbf{c}_i\}_{i \leq N_c}$, where $N_c = \lceil \frac{N}{C} \rceil$. That is, $\mathbf{c}_i = \mathbf{x}_{i,:}$ where $\mathbf{x}_{i,j} = x_{C \cdot i + j}$ (numpy indexing notation). CAT compresses each chunk $\mathbf{c}_i = \mathbf{x}_{i,:}$ using the *compressor* $f_\theta$ into chunk representations: $f_\theta(\mathbf{c}_i) \in \mathbb{R}^{D_g}$.

$$\mathbf{x} = \{x_1, \cdots x_N\} \xrightarrow{\text{chunking}} \{\mathbf{c}_1, \cdots \mathbf{c}_{N_c}\} \xrightarrow{\text{compress}} \{f_\theta(\mathbf{c}_1), \cdots f_\theta(\mathbf{c}_{N_c})\}.$$

After compression, CAT decodes the original sequence from the compressed chunk representations $\{f_\theta(\mathbf{c}_i)\}$. The *decoder* $g_\theta$ takes in compressed chunk representations of the past tokens as input and outputs a distribution over the tokens in the next chunk. Formally, the *decoder*'s predictive distribution for the tokens in $i$th chunk $\mathbf{c}_i$ is

$$p_\theta(\mathbf{c}_i \mid \mathbf{c}_{i-1} \cdots \mathbf{c}_1) = \prod_{j=1}^{C} g_\theta\big( \underbrace{\mathbf{x}_{i,j}}_{j^{\text{th}} \text{ token in chunk } \mathbf{c}_i} \mid \underbrace{\mathbf{x}_{i,j-1}, \dots \mathbf{x}_{i,1}}_{\text{previous tokens in chunk } \mathbf{c}_i}, \underbrace{f_\theta(\mathbf{c}_{i-1}) \cdots f_\theta(\mathbf{c}_1)}_{\text{past chunk representations}} \big) \quad (1)$$

By using compressed chunk representations, CAT reduces the amount of compute and memory required for decoding; the larger the chunk size the larger the reduction in memory and compute.

During training, the compression and the decoding happens in parallel for all tokens in the sequence because compression of a chunk does not depend on an earlier chunk. This choice allows entire CAT model to be *efficiently* trained end-to-end with the standard next-token prediction loss. The end-to-end training ensures that CATs *learn what to retain* in their compressed chunk representations rather than relying on fixed attention patterns, or complex state update rules.

**Training for test-time flexibility in compute and memory.** Varying the chunk size in CATs trades-off quality for compute and memory efficiency. Training CATs with multiple chunk sizes during training renders a single adaptive model whose compute-memory budget can be adjusted directly at test-time without any retraining.

To build such a controllably efficient CAT model, we uniformly sample a chunk size $C$ at each training iteration, and pass in a *learnable* indicator token to CAT to indicate which chunk size it is currently operating at. The compressed tokens are separated from the uncompressed ones in the decoder using a marker token shared across different chunk sizes. After training, one can use the same CAT model at different compute/memory budget at test-time by just changing the indicator token. Appendix B.4 provides further detail.

### 2.1 How to implement fast and scalable CATs

Due to both components of CAT being transformers, CAT admits a pure PyTorch efficient implementation for scalable training and fast generation. We describe the approach here.

**Training.** While CATs are simple and build on dense transformer abstractions, their *naive PyTorch training implementation is very inefficient*. Note that compression of chunks of tokens is efficient since it can be done in parallel, specifically using `torch.vmap(`$f_\theta(\mathbf{c}_i)$`)` for all chunks $\mathbf{c}_i$. This costs a total of $O(\frac{N}{C} \cdot C^2) = O(NC)$ in self-attention compute, which is much better than $O(N^2)$. But, computing logits for tokens in chunk $\mathbf{c}_i$, that is computing $g_\theta(\mathbf{c}_i \mid f_\theta(\mathbf{c}_1) \cdots f_\theta(\mathbf{c}_{i-1}))$ can be non-trivial since for chunk $\mathbf{c}_i$, we have $i-1$ past chunk representations $\{f_\theta(\mathbf{c}_1), f_\theta(\mathbf{c}_2) \ldots f_\theta(\mathbf{c}_{i-1})\}$. In other words, there are different number of past chunk representations for every chunk, making shapes variable and as a result, harder to parallelize computation of logits. One could employ a python loop and compute logits for every chunk sequentially, but that would be slow and won't scale. In fact, even if one manages to compute logits for every chunk in parallel, the total self-attention operations in the decoder would be $O(\sum_{i=1}^{N_c}(i+C)^2) = O((\frac{N}{C})^3)$, that is cubic in sequence length. Padding to make shapes constant would make things worse. Thus, naive techniques will not scale, despite CATs being a simple architecture. Similar architectures (Ho et al., 2024; Yu et al., 2023) do not have this problem: computing logits can be naively parallelized due to fixed shapes and self-attention operations scale quadratically due to a single compressed representation of the past.

Now, in CATs, observe that in computing logits for every chunk $\mathbf{c}_i, \mathbf{c}_{i+1} \ldots \mathbf{c}_{N/C}$, one calculates exactly the same key-value vectors for the representation $f_\theta(\mathbf{c}_j)$ in the decoder transformer, where $j < i$. This points to repeated and identical computations. We exploit this observation in CATs making the training scalable This way of computing logits is quadratic in sequence length but a constant times better: $O(\frac{N^2}{C})$ vs. the $O(N^2)$ complexity of the dense transformer.

On a high-level, we implement this by modifying the original sequence $\mathbf{x} = \{\mathbf{c}_1, \ldots \mathbf{c}_i \ldots\}$ to $\{\mathbf{c}_1, f_\theta(\mathbf{c}_1), \mathbf{c}_2, f_\theta(\mathbf{c}_2), \ldots \mathbf{c}_i, f_\theta(\mathbf{c}_i) \ldots\}$, that is we insert compressed representations of the chunk after the chunk of tokens itself. Now, we pass this sequence into the decoder during training, with a custom attention mask (Figure 9) that allows a token in chunk $\mathbf{c}_i$ to attend to previous tokens within that chunk and *only* to previous chunk representations, which would be $f_\theta(\mathbf{c}_{i-1}), f_\theta(\mathbf{c}_{i-2}) \ldots f_\theta(\mathbf{c}_1)$. Any token in chunk $\mathbf{c}_i$ does not attend to raw tokens outside this chunk. This implementation allows re-use of key-values for chunk representations $f_\theta(\mathbf{c}_i)$ in decoder for computing logits of a future chunk $\mathbf{c}_j$, where $j > i$.

**Generation.** The decoder during generation attends to atmost $N_c + C$ tokens. Due to compression, CATs can throwaway past chunks of tokens, and only keep their compressed chunk representations in memory. This straightaway results in a big reduction of memory; the KV cache is slashed by a factor of $C$. For even a moderate chunk size of 4, this results in big reductions in memory during generation (Figure 2). This slash in memory is accompanied by reduced memory accesses the decoder makes in CATs, which is the major bottleneck during generation. The decoder attends to atmost $N_c + C$ tokens during generation, reducing compute required in self-attention significantly.

Implementing generation is simpler than training and very similar to how it occurs for a dense transformer. In fact, a pure PyTorch implementation[3] for CATs is on-par with efficient architectures that utilize custom kernels. Given a sequence, CATs first compute representations for each chunk

---

[3]Our implementation is inspired by the gpt-fast code.

in parallel and use them to prefill the decoder's KV cache. Then generation proceeds chunk by chunk: each new chunk is decoded token by token in the decoder, and once a chunk is complete, the chunk is compressed and its representation is prefilled in the KV cache for the generation of the next chunk. This loop continues until the sequence is fully generated. The full implementation details are in App. D and D.3, refer to App. B for a PyTorch style pseudo-code.

## 3 RELATED WORK

**Efficient self-attention:**  Many works have proposed techniques to reduce inference costs of self-attention. These techniques include *heuristically* defined *fixed* sparse or stratified attention masks Child et al. (2019); Zaheer et al. (2020) or local sliding window masks Jiang et al. (2023) that reduce the tokens being attended to in self-attention. The compute required (and in some attention masks, memory) for attention go down during generation, but if the *wrong* attention mask is chosen for the task, these methods will be less performant or will require more depth (Arora et al., 2024a). To match quality of a dense transformer, these models either require big window sizes (making their memory costs large again) or need to be composed with dense attention again at specific layers (Arora et al., 2024a; Agarwal et al., 2025).

Instead of reducing inference costs through efficient attention, Mixture-of-Experts (Shazeer et al., 2017; Dai et al., 2024) take a different approach where they increase parameters without increasing inference costs in attention. Having said that, since MoEs increase parameters in the feed-forward layers, they are complementary to every sequence mixers including the ones present inside of CAT (either in decoder or compressor).

**Compressing past context:**  Rae et al. (2020); Chevalier et al. (2023) explored recurrent formulations of a transformer to enable generation of longer sequences on limited compute and memory by compressing past context. But sequential training is slow and memory intensive, making these approaches hard to scale on modern hardware that favors parallel computations. Moreover, training models in a recurrent fashion has optimization challenges, back-propagation through time (BPTT) being the most important one. More recently Geiping et al. (2025) had to use very careful recipe to train a large recurrent architecture in a stable manner and prevent optimization collapse.

Alternatively, Native Sparse Attention (NSA) (Yuan et al., 2025) reduce attention compute by attending to compressed chunks of tokens as well as to specific chunks of uncompressed tokens in the past. These past tokens are compressed in parallel in every layer. This is similar in spirit to our work, however there are no memory savings during inference since the KV cache needs to be retained for the entire past context; there are only compute savings.

**Linear attention:**  Arora et al. (2024a); Katharopoulos et al. (2020) linearize self-attention that replace softmax-based attention with kernelized dot-product-based linear attention, that further admits a linear recurrence form. Recent enhancements incorporate data-dependent gating mechanism in the recurrence (Dao & Gu, 2024; Yang et al., 2025b) all which require *handcrafted* and *complicated* recurrent state update rules. Although these architectures show impressive reductions in compute and memory, the fixed-size recurrent state struggles to manage information over long sequences, that hurts in-context recall performance (Arora et al., 2024a; Jelassi et al., 2024; Wen et al., 2024). To make these mixers competitive, they are usually composed with long sliding window attention at specific layers (Yang et al., 2025b). Performing such a composition is unclear and requires careful *trial-and-error* (Waleffe et al., 2024; Qwen, 2025) making the design process for an efficient architecture highly cumbersome.

**Hierarchical transformers:**  Nawrot et al. (2021; 2022); Slagle (2024) explored downsample-then-upsample approach (*hour-glass* like structure), where the sequence is downsampled into *coarse* tokens followed by upsampling into *fine-grained* tokens before being decoded. Due to the *hour-glass* structure, there are compute savings during training; but the architecture must maintain a cache for all the past tokens leading to significant memory accesses (especially for *fine-grained* ones) which is the main bottleneck during generation.

Unlike the above, Ho et al. (2024); Yu et al. (2023) break up the modeling of a sequence into independent chunks/patches, given a single compressed representation of the entire past. While compression helps in efficiency, the requirement to decode each chunk from a fixed size compressed

| Method | Unrestricted Access to Memory? | Flexible memory? | Scalable training? | Compute & memory efficient? | Adaptive? |
|---|---|---|---|---|---|
| ***Dense Attention***: Vaswani et al. (2017) | ✓ | ✓ | ✓ | ✗ | ✗ |
| ***Sparse Attention***: Child et al. (2019) | ✗ | ✓ | ✓ | ✓ | ✗ |
| ***NSA***: Yuan et al. (2025) | ✓ | ✓ | ✓ | ✗ | ✗ |
| ***Sliding window Attn.***: Jiang et al. (2023) | ✗ | ✗ | ✓ | ✓ | ✗ |
| ***Linear Attention***: Dao & Gu (2024) | ✓ | ✗ | ✓ | ✓ | ✗ |
| ***Recursive compression***: Chevalier et al. (2023) | ✓ | ✓ | ✗ | ✓ | ✗ |
| ***MegaByte/Block Transformer***: Ho et al. (2024); Yu et al. (2023) | ✓ | ✗ | ✓ | ✓ | ✗ |
| ***CATs*** | ✓ | ✓ | ✓ | ✓ | ✓ |

Table 1: We categorize the existing related work into key properties that are desirable for an efficient architecture. *"Both compute and memory efficient?"* signifies savings during inference; *"Unrestricted Access to Memory"* signifies whether an architecture can freely access any part of the memory in the past, without any restrictions. We provide a discussion in Sec. 3 and an extended discussion in App. E

representation results in poor in-context recall even on simple toy tasks (App. Fig. 3). Further, unlike the original encoder-decoder architectures that attend directly to past tokens (Raffel et al., 2020; Vaswani et al., 2017), decoder in CAT attends to the compressed representations of chunks of tokens in the past.

CATs sidestep many limitations of existing efficient baselines described above. Firstly, CATs are *simple*: they do not require any handcrafted state update rules or careful composition with attention layers to have competitive performance; CATs directly build on dense transformer abstractions. Secondly, CATs alleviate the fixed memory by having flexible but efficient memory usage. That is the memory grows *gracefully* as sequence length increases, resulting in superior in-context recall performance, despite using similar memory overall compared to fixed memory baselines (Table 2). Thirdly, CATs have scalable and efficient training where compression and decoding can happen in parallel. Finally, CATs allow control of quality-compute trade-offs at test-time, allowing them to cater to downstream tasks with different budgets. This is similar in spirit to Kusupati et al. (2022); Devvrit et al. (2023); Beyer et al. (2023).

We provide a brief summary of the related work in Table 1, indicating key properties where CATs and other methods differ. For an extended related work, refer to Appendix E.

## 4 EXPERIMENTS

### 4.1 EVALUATION CRITERIA

We first discuss and motivate what the right evaluation criteria among different models should be: all having potentially heterogeneous and diverse architectural components, from a practical language model deployment point of view.

**Inference costs dominate.** Major costs of language models include (i) training these models, and then (ii) serving these models for inference. While both training and inference costs of these models are significant and important, the cost of inference *dominates* in the long run when deploying language models in the real world (Timbers, 2023; Jassy, 2023; Monetizely, 2025; Bridi, 2025; GMI Cloud, 2025; Dhavle, 2025). In fact, the costs of training can break even with inference in *just* a few weeks after deployment (Timbers, 2023). Further, this time to break even is decreasing rapidly

as more people are using reasoning language models with increasing context lengths (Guo et al., 2025).

**Raw parameter counts do not impact inference costs.** Now, inference costs depend on the time that a model takes to process/generate a sequence (computations required to perform forward passes) and the hardware/GPUs required to process/generate sequences (the memory required to run the model). **The raw parameter count affects the inference cost only insofar as it affects computations required and memory usage.** As an example, consider Mixture-of-Experts (MoEs) Shazeer et al. (2017), which are now the default choice in all flagship LLMs (Yang et al., 2025a; Agarwal et al., 2025). MoEs are *explicitly* designed to have more parameters ($10 - 20\times$ more (Yang et al., 2025a; Agarwal et al., 2025)) than a dense transformer while keeping inference costs essentially unchanged (Shazeer et al., 2017).

**As a result, what matters for real-world deployment is what quality a model obtains at a desired inference cost, and not the raw parameter count itself.**

### 4.2 BASELINES AND TRAINING SETUP

**Baselines:** Our experiments provide a comprehensive comparison of recent state-of-the-art architectures, including (i) attention-based baselines: standard Dense Transformer (Touvron et al., 2023) and Sparse Transformer (Child et al., 2019), (ii) Linear Transformers such as Mamba2 (Dao & Gu, 2024) and GatedDeltaNet (GDN) (Yang et al., 2025b), as well as (iii) hybrid architectures such as the hybrid variant of GDN having alternate layers as long sliding windows, GDN-Hybrid (GDN-H-1:1). All models use $L = 12$ layers with hidden size of $D = 1024$. GDN-Hybrid employs a sliding window of 2K, following Yang et al. (2025b).

While we picked standard baselines from the literature (Yang et al., 2025b; Dao & Gu, 2024) at their default hyperparameter settings, to aid in further fair evaluation of different models and prevent any cherry-picking, we further **scale up each model type** by changing their model configuration. Specifically, we add scaled up version of GDN by increasing the recurrent state size (GDN-2X); we change linear-dense attention ratio in GDN-Hybrid to 1:7 and change the attention to a global attention (GDN-H-1:7-GLOBAL), and further scale up the model dimension to $2\times$ (GDN-H-1:7-GLOBAL 2D). Further, we add GDN-Hybrid with ratios 1:4 with 2K sliding windows, to add additional configuration for GDN-Hybrid 1:1 model. To enable more points for comparison, we decrease the hidden size in Dense (Dense-D/2) and GDN-H-1:1 (GDN-H-1:1 D/2) by $2\times$. Finally, we decrease the sparsity ratio in Sparse to half and increase model dimension by $2\times$ (Sparse-4).

**This process results in a total of 12 models, all have different architectural components (dense attention, linear attention, hybrids), with a range of parameter counts from $\sim$300M to upto $\sim$820M, with varying inference costs. Every model type has atleast $\geq 2$ model configurations to ensure fair and broader evaluation. We compare these 12 models against a *single* CAT model.**

Refer to Appendix D for more details regarding hyperparameters used for each model.

**What makes CATs purr?** To match dense-transformer perplexity, we empirically find a more *expressive* decoder helps: that is, decoder uses $2\times$ hidden size. *This suggests accurate decoding from compressed representations needs higher dimensionality*, with similar observations in recent works (Ho et al., 2024; Yu et al., 2023). Refer to App. C.2 for a comparison. Further, we find depth of compressor does not have major effect on perplexity (App. C). Given these findings, to instantiate CATs that compete with dense transformer of depth $L$ and hidden size $D$: CATs use a decoder of depth $L$ and hidden size $2D$, and a compressor of depth $L/4$ and hidden size $D$. While this increases parameters, CATs are **still** significantly faster and memory efficient compared to the corresponding dense transformer, similar to MoEs (Shazeer et al., 2017). Thus, for CATs we use $L = 12$ layers, same as baselines, but a wider hidden size of $D_g = 2D = 2048$ for the decoder. The compressor uses $L = 3$ layers and hidden size of $D_f = D = 1024$. This makes the parameter count for CATs close to $1B$. We train CATs simultaneously on chunk sizes $C = \{4, 8, 16, 32\}$. *Note that this CAT is a single model that can work with different chunk sizes at once, offering different compute-quality trade-offs at test-time.*

**Training setup:** All models were trained on 15B tokens of FineWeb-Edu Penedo et al. (2024) which is $2.5\times$ the Chinchilla optimal, with a context length of 4K following Behrouz et al. (2024); Yang

et al. (2025b). We use the AdamW optimizer Loshchilov & Hutter (2017) with a peak learning rate of 8e-4, weight decay of 0.1, gradient clipping of 1.0, batch-size of 0.5M tokens, employing the GPT2 tokenizer (see Appendix D for more details).

## 4.3 RESULTS

Keeping the discussion in Section 4.1 in mind, **we compare different models by measuring what quality they get for a given inference cost**, specifically memory usage, since it is the major inference time bottleneck in increasingly memory-bound GPU workloads (Gholami et al., 2024). We measure quality using real-world in-context recall tasks. Figure 1a reports these results. We report quality at a given latency requirement in App. Figure 5.

**Language modeling and understanding benchmarks:** Table 5 in appendix reports the zero-shot perplexity against LAMBADA (LMB) Paperno et al. (2016), WikiText (Wiki) Merity et al. (2016), and on a held-out test set of FineWeb-Edu (FW), and the zero-shot accuracies on key common-sense reasoning benchmarks; Appendix D.2 expands the acronyms in table 5. All CAT variants outperform existing efficient baselines except one, on common-sense reasoning benchmarks. These evaluations however only consider short sequences ($\leq$ 30 tokens on average). Further, note that perplexity **does not** correlate with downstream long-context performance (Fang et al., 2025). Hence, we test language understanding on longer contexts in Table 3 on a suite of tasks from LongBench (Bai et al., 2023) and test in-context recall on real world tasks (Arora et al., 2023b) (Table 2) where CAT outperforms all models when appropriately inference costs matched, all using a single model. In Tables 3 and 5, we only report models that obtain competitive language modeling to the dense transformer (Dense D).

**Real world in-context recall:** Table 2 reports results on in-context recall tasks from Arora et al. (2024a). **Figure 1a reports these results when models are appropriately inference costs matched.** We report results on SWDE and FDA, which have longer sequences among the datasets in the suite (others have an average length of $\leq$ 300 tokens (Arora et al., 2024b)). Appendix A shows evaluations on all datasets. Linear models (Mamba2, GatedDeltaNet) lag far behind dense attention, while GDN-Hybrid reduces the gap. CAT surpasses nearly all efficient baselines, benefiting from the gracefully growing memory. CAT outperforms even the dense transformer at moderate chunk sizes ($= 4, 8$), while being at least $1.4\times$ faster and $2.2\times$ more memory efficient (see appendix A.5). We additionally provide how in-context recall trades-off with latency in Appendix Figure 5. More details can be found in Appendix D.

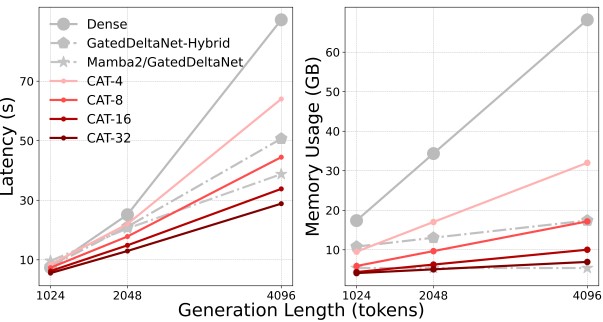

| Model | SWDE | FDA | Avg. |
|---|---|---|---|
| Dense | 43.4 | 19.7 | 32.0 |
| Dense-D/2 | 32.0 | 22.0 | 27.0 |
| Sparse-4 | 36.0 | 16.0 | 26.0 |
| Sparse-8 | 20.9 | 6.0 | 13.0 |
| Mamba2 | 13.5 | 4.5 | 9.0 |
| GDN | 18.0 | 6.8 | 12.0 |
| GDN-2× | 24.0 | 11.0 | 17.5 |
| GDN-H 1:1 | 44.0 | 17.8 | 31.0 |
| GDN-H 1:1 D/2 | 17.0 | 3.2 | 10.0 |
| GDN-H 1:4 | 33.0 | 20.0 | 27.0 |
| GDN-H 1:7 G | 38.0 | 25.0 | 32.0 |
| GDN-H 1:7 G 2D | 46.0 | 36.0 | 41.0 |
| CAT-4 | **49.1** | **45.1** | **47.1** |
| CAT-8 | 38.2 | 34.8 | 36.5 |
| CAT-16 | 27.5 | 15.4 | 21.5 |
| CAT-32 | 13.2 | 3.2 | 8.2 |

Figure 2: A single CAT model generates $1.4 - 3.2\times$ faster than the dense transformer while showcasing upto $2.2 - 9.5\times$ lower memory usage.

Table 2: Zero-shot performance on real-world in-context recall tasks measured upto 4K sequence lengths. H and G stand for Hybrid and Global respectively. **Fig. 1a gives an inference costs matched comparison.**

**Needle-in-haystack & State-tracking:** Table 4 reports results on RULER Hsieh et al. (2024) single-needle tasks: S-NIAH-N (recall number from the context). Linear recurrent models (Mamba2, GDN) struggle at longer contexts, and while GDN-Hybrid narrows the gap with dense transformers, performance still drops at longer contexts. CATs-4/8/16 outperform the efficient baselines as context length increases, showing slower degradation with length,

even compared to the dense transformer. This slow degradation can possibly be attributed to reduced sequence length in CAT that leads to fewer distractions for attention (Barbero et al., 2024; Chiang & Cholak, 2022; Golovneva et al., 2025). Further, large-chunk CAT underperforms at short contexts but interestingly surpass baselines at long ones. One reason why large-chunk CAT underperforms could be due to ineffective compression – due to larger chunks, the compressor in CAT is not always able to surface the *right* information in the chunk representation for accurate retrieval. More pre-training or finetuning on specific task data alleviates this problem for large chunk CAT (see App. A.13). That being said, note that there is an upper limit to how much information fixed sized chunk representations can practically learn to hold for large token chunks. Table 11 reports results on the harder S-NIAH-U (recall a long alpha-numeric string or UUID) and BabiLong state-tracking task (qa1 subset). On the BabiLong task, all models decline as context grows, although linear recurrent models (Mamba2, GDN) perform better, in accordance with Kuratov et al. (2024).

**Benchmarking generation:** Figure 2 compares architectures as one scales the sequence length, with a fixed batch-size of 320 to maximize throughput. CAT generates sequences $1.4 - 3.2\times$ **faster** than the dense transformer while showcasing **upto** $2.2 - 9.5\times$ **lower total memory usage** as one increases chunk sizes, despite using significantly more parameters than the baselines due to wider decoder and the additional compressor. This is not surprising since the major bottlenecks during generation are: (a) KV cache size that drives the main memory requirement during generation and not the parameter count (Sec. 5), (b) memory accesses required for a token, and (c) FLOPs used per token determined by the past tokens being attended to. CATs reduce these factors despite carrying more parameters overall. See appendix D.3 for implementation details.

| | Single-doc QA | | Multi-doc QA | | Few Shot | | Avg. |
|---|---|---|---|---|---|---|---|
| **Model** | QAS | MQA | HQA | 2WMQ | TQA | TREC | |
| Dense | 3.9 | 12.2 | 6.9 | **10.8** | 11.2 | 10.6 | 9.3 |
| Sparse | 5.1 | 11.0 | 7.0 | 10.6 | 10.5 | 5.6 | 9.3 |
| Mamba2 | 4.1 | 11.9 | **7.6** | 7.6 | 9.0 | 7.6 | 8.0 |
| GDN | **8.3** | **15.5** | 6.0 | 7.9 | 7.4 | 8.3 | 8.9 |
| GDN-2× | 4.1 | 11.8 | 6.7 | 9.6 | 9.8 | 6.8 | 8.1 |
| GDN-H 1:1 | 4.2 | 13.3 | 6.6 | 11.6 | 11.8 | 6.5 | 9.0 |
| GDN-H 1:4 | 4.6 | 13.0 | 7.0 | 10.4 | 10.6 | 24.2 | 11.6 |
| GDN-H 1:7 G | 4.7 | 12.5 | 6.4 | 12.2 | 9.2 | 28.3 | 12.2 |
| GDN-H 1:7 G 2D | 3.9 | 12.9 | 8.3 | 9.1 | 11.9 | 23.7 | 11.6 |
| CAT-4 | 5.6 | 12.7 | 7.4 | 9.9 | 12.1 | **35.6** | **13.9** |
| CAT-8 | 5.5 | 11.0 | 6.1 | 8.0 | **12.4** | 29.5 | 12.1 |
| CAT-16 | 4.3 | 14.1 | 6.1 | 5.6 | 10.5 | 16.6 | 9.5 |
| CAT-32 | 4.7 | 11.0 | 7.0 | 6.6 | 10.0 | 8.3 | 7.9 |

Table 3: Zero-shot evaluation of baselines on suite of tasks from LongBench Bai et al. (2023) up to 4K tokens. Refer to Appendix D.2. CAT-4/8/16/32 are a single model.

**Instantiating CAT as a layer:** While CAT presented in the paper is a separate architecture, one can take the core concepts and instantiate CAT as a layer that can be swapped in any sequence model as a drop-in replacement. This can unlock lots of interesting possibilities starting with creating hybrid as well as adaptive architectures that mixes CAT layers alongside dense attention, or perhaps even linear attention. We leave this open for future work. App. A.7 reports results when instantiating CAT as a layer on MQAR task along with more details.

Table 4: Accuracy on RULER Hsieh et al. (2024) S-NIAH-N benchmark.

| | S-NIAH-N | | |
|---|---|---|---|
| **Model** | **1K** | **2K** | **4K** |
| Dense | 96.0 | 92.0 | 43.0 |
| Sparse | 51.2 | 46.2 | 5.0 |
| Mamba2 | 97.7 | 81.1 | 18.6 |
| GDN | 84.7 | 69.1 | 13.6 |
| GDN-2× | 78.0 | 61.4 | 29.0 |
| GDN-H 1:1 | 99.0 | 97.0 | 44.0 |
| GDN-H 1:4 | 98.0 | 96.0 | 35.8 |
| GDN-H 1:7 G | 98.3 | 93.0 | 23.2 |
| GDN-H 1:7 G 2D | **99.5** | **99.3** | 43.8 |
| CAT-4 | 96.0 | 97.0 | **96.0** |
| CAT-8 | 90.0 | 93.0 | 91.0 |
| CAT-16 | 76.0 | 72.0 | 70.0 |
| CAT-32 | 60.0 | 37.0 | 31.0 |

**CAT is a *meta* sequence mixer**: CAT has two components: a compressor and a decoder – each of these could make use of any sequence mixers, such as linear attention. Appendix A.8 provides preliminary result on the MQAR task where the decoder in CAT is a GDN-HYBRID architecture (Yang et al., 2025b) (having a 1:1 dense-to-linear ratio). This new architecture solves this task, empirically demonstrating the use of linear attention layers inside of CAT: meaning rather than CAT being a strict competitor to linear attention (or any other efficient sequence mixer), **CAT is complementary**. Further, the use of a different sequence mixers inside of CAT can unlock the test-time control of efficiency with those sequence mixers (e.g., GDN in this case). We use the same setup described in Appendix A.6.

**CATs scale similar to dense transformers while still having lower inference costs:** We provide a scaling experiment in appendix Figure 6.

**Ablations:** We investigate how different choices affect performance of CATs in App. C.

## 5 DISCUSSION AND CONCLUSION

We introduce **C**ompress & **A**ttend **T**ransformers (CATs), a simple *controllably* efficient alternative to the standard transformer architecture. A single adaptive CAT model achieves a pareto frontier in quality-inference costs curve, outperforming every popular architecture **across 12 models** with different model configurations, parameter counts ($\sim$300M to $\sim$820M) and varying inference costs on real-world in-context recall tasks **when appropriately inference costs (memory) matched** (see Figure 1a). Notably, CAT-4 (the least efficient setting) outperforms the dense transformer in both language modeling and recall tasks while being $1.5\times$ faster and requiring $2\times$ less memory. We discuss a possible explanation for this observation, followed by the practical utility of CATs and some future directions.

**On Parameters and Inference Costs.** As discussed in Section 4.1, parameters do not *directly* affect inference costs. Despite the larger parameter count in CATs, working with a reduced sequence due to compression ensures that CATs have similar or lower inference costs than their dense transformer equivalents. In spirit, CATs are similar to flagship models that are all parameterized as Mixture-of-Experts (MoEs) (Shazeer et al., 2017), where increased parameter counts in MoEs (upto $10\times$ more[4]) does not mean higher inference costs. In fact, more parameters brings improvement in quality while using the similar or lower inference cost (Dai et al., 2024). **Due to compression, CATs utilize their increased model parameters *smartly***, bringing improvements in quality (in-context recall) while still being efficient, similar to recent trends in literature (Li et al., 2024a).

**Are CATs adoptable?** Current implementation for training CATs costs twice as much time due to inefficient compilation (using recent PyTorch FlexAttention API (Dong et al., 2024)) of attention mask employed in CATs (see Appendix B.5 for a discussion); custom kernels can be developed in future to mitigate this difference and potentially realize compute savings during pre-training discussed in Section 2.1. Further, CAT amortizes this overhead in training by effectively training multiple models in a single pre-training run (corresponding to different inference budgets). However, training is a one time cost, and the service life of models dictates profits, making *serving costs the more important consideration*. Note that deploying language models at scale is often constrained *not* by model weights but by the memory footprint of their KV cache (**inference costs**). For instance, `Qwen3-14B` at a modest batch size of 16, which is common in chat/code completion, requires an *order of magnitude more memory* for the KV cache than the model weights themselves: 28GB for the weights vs. $\sim$ 670GB for the KV cache at maximum context length. In contrast, a CAT variant of the same model could reduce total memory usage upto $\sim$ $4\times$ despite having more model parameters overall[5], and lead to higher generation throughput. As most modern GPU workloads are increasingly memory-bound rather than compute-bound, memory reductions play an even more critical role (Gholami et al., 2024). The reduction in memory and increase in throughput is more pronounced at larger batch sizes for CATs, which are critical for workloads such as synthetic data generation (Maini et al., 2025) and large-scale rollouts in reinforcement learning (RL) post-training pipelines for math/code reasoning or alignment for faster RL training (Noukhovitch et al., 2024) or better RL optimization (Zhang & Ranganath, 2025). Further, note that CATs serve as multiple models in one, enabling reduced compute during high traffic, longer shelf-life under smaller budgets, and deployment on cheaper hardware – **all from a single training run**.

**Future work:** As shown in Section 4, CAT is complementary to existing sequence mixers, and developing new hybrids is an interesting direction: for e.g. linear attention as *compressor* with dense attention *decoders* for long-range interactions between the compressed sequence. A different direction is data-dependent adaptivity. CATs, as they stand, require users to choose a chunk size appropriate for their compute and memory budgets. Instead, one could post-train with RL to allow CATs to learn to allocate budget themselves based on the context and the task. Such post-training would enable truly adaptive efficiency. Further, as illustrated before, CATs could possibly be instantiated as a layer instead as a full architecture, where every layer has a separate compressor and decoder. This could enable interesting hybrid and adaptive architectures combining dense attention layers mixed with CAT layers. Additionally, one can train a CAT where compression only kicks in only after 1K tokens (say), resulting in parallel compression of older context only. Finally, scaling up CATs to larger model scales and longer training would enable further insights and better comparisons.

---

[4]`Qwen3-30B-A3B`: only 3B parameters are *active* during inference out of 30B parameters in total

[5]Total memory usage for CATs: $28 \cdot \left(4 + \frac{1}{4}\right) + \frac{670 \cdot 2}{32} = 160$GB, which is $\sim 4\times$ better at chunk size $C = 32$

## 6 REPRODUCIBILITY STATEMENT

We provide exhaustive implementation details for CATs in Section 2.1 and pseudo-code in Appendix B. Further, we provide training details and hyperparameters for baselines in Appendix D. We directly use the official code for implementing and benchmarking baselines.

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

TABLE OF CONTENTS FOR THE APPENDICES

## A MORE EXPERIMENTS

### A.1 LANGUAGE MODELING EVALUATIONS

| Model | LMB↓ | Wiki↓ | FW↓ | HS↑ | PQ↑ | AE↑ | AC↑ | WG↑ | OQA↑ | Avg.↑ |
|---|---|---|---|---|---|---|---|---|---|---|
| Dense | 38.7 | 19.6 | 17.1 | 34.8 | 65.6 | 56.7 | 24.4 | 51.1 | 20.0 | 42.1 |
| Sparse-8 | 37.2 | 18.5 | 16.0 | 35.6 | 66.8 | 57.3 | 25.4 | 51.1 | 22.8 | 43.2 |
| Mamba2 | 36.1 | 19.5 | 16.7 | 36.1 | 67.0 | 59.2 | 26.5 | 51.9 | 21.6 | 43.7 |
| GDN | 35.7 | 18.8 | 16.3 | 36.1 | 66.8 | 58.7 | 25.2 | 51.6 | 22.8 | 43.5 |
| GDN-2× | 35.3 | 18.4 | 16.0 | 35.9 | 67.4 | 58.6 | 27.2 | 51.8 | 21.8 | 43.8 |
| GDN-H 1:1 | 36.6 | 18.5 | 16.2 | 36.8 | 66.3 | 56.4 | 25.8 | 52.1 | 20.4 | 43.0 |
| GDN-H 1:4 | 36.0 | 21.4 | 17.4 | 34.8 | 67.0 | 57.0 | 26.5 | 50.3 | 22.0 | 42.9 |
| GDN-H 1:7 G | 35.8 | 18.4 | 16.1 | 36.0 | 67.6 | 57.4 | 26.6 | 51.5 | 23.4 | 43.7 |
| GDN-H 1:7 G 2D | 31.2 | 15.1 | 13.6 | 38.6 | 70.1 | 62.3 | 27.8 | 52.7 | 23.6 | 45.8 |
| CAT-4 | 38.0 | 18.1 | 16.0 | 35.6 | 66.4 | 59.5 | 27.1 | 51.5 | 23.4 | 43.9 |
| CAT-8 | 37.2 | 18.1 | 15.8 | 35.4 | 66.8 | 60.1 | 27.4 | 51.3 | 23.6 | 44.1 |
| CAT-16 | 36.8 | 18.4 | 16.0 | 35.5 | 67.3 | 60.2 | 27.0 | 52.0 | 23.8 | 44.3 |
| CAT-32 | 36.8 | 19.1 | 16.4 | 35.9 | 68.2 | 61.0 | 27.0 | 53.6 | 25.0 | 45.1 |

Table 5: Zero-shot perplexity and accuracy on language modeling and common-sense reasoning benchmarks. CAT outperforms all models on common-sense reasoning and language modeling evaluations except one. However, note that these evaluations considers short sequences only (≤ 30 tokens on average). Further, note that perplexity **does not** correlate with downstream long-context performance (Fang et al., 2025). Hence, we test language understanding on longer contexts in Table 3 on a suite of tasks from LongBench (Bai et al., 2023) and test in-context recall on real world tasks (Arora et al., 2023b) (Table 2) where CAT outperforms all models when appropriately inference costs matched, all using a single model. In Tables 3 and 5, we only report models that obtain competitive language modeling to the dense transformer (Dense).

### A.2 CATS OUTPERFORM BASELINES WHEN MEMORY MATCHED

To rule out any memory discrepancy, (Fig. 2) evaluates on MQAR task (Arora et al., 2023a), matching memory budgets down to the level of bytes, and stress-tests models up to 1K sequence length (5× standard); Figure 4 in reports results. Baselines are grid-searched over learning rates. Linear models collapse at longer contexts, while CATs remain near-perfect, thanks to the flexible yet efficient memory scaling. We use the same setup in App. A.6.

### A.3 COMPARISON WITH MEGABYTE/BLOCK TRANSFORMER

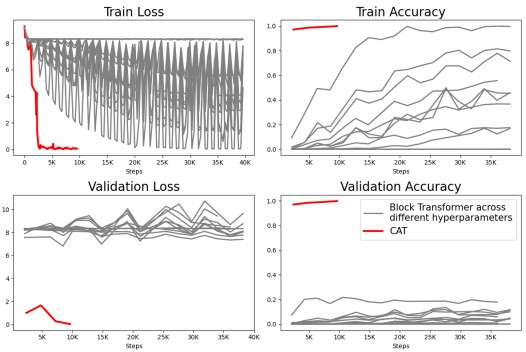
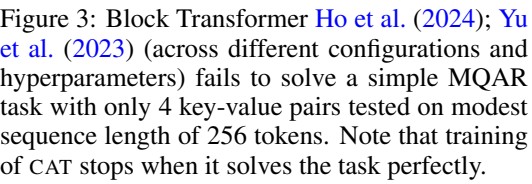
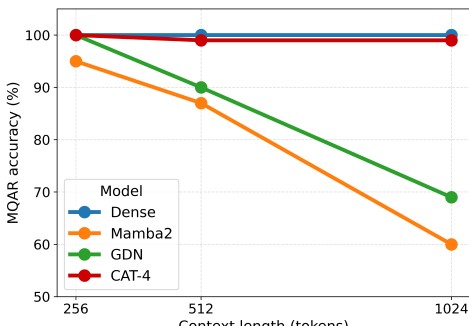

Figure 3: Block Transformer Ho et al. (2024); Yu et al. (2023) (across different configurations and hyperparameters) fails to solve a simple MQAR task with only 4 key-value pairs tested on modest sequence length of 256 tokens. Note that training of CAT stops when it solves the task perfectly.

Figure 4: Comparison of different architectures across sequence lengths on MQAR task. We measure test-accuracy on the hardest subset. All architectures are memory matched in bytes at every point (except dense transformer).

The MegaByte/Block Transformer (Ho et al., 2024; Yu et al., 2023) has elements similar to CAT but fail to solve a simple in-context recall task in fig. 3 across different hyperparameters and architecture configurations due to the fixed memory bottleneck. In fact, the block transformer overfits on the task. CATs alleviate the memory bottleneck with a gracefully growing memory, allowing it to solve the task, with even lower memory requirements.

In figure 3, we evaluate in-context recall ability for Block Transformer architectures Ho et al. (2024); Yu et al. (2023), that model chunks of tokens similar to CATs but with a subtle but salient difference in the architecture circuit (that we explain below). For this experiment, we test on the MQAR task (a synthetic needle-in-haystack task Arora et al. (2023a)) on a modest sequence length of 256. We test the accuracy of retrieving just 4 needles. We parametrize components of Block Transformer that is: global model and local model using a transformer, the embedder is a look-up table or a transformer. We keep the patch size/chunk size as 4 – same as CAT. We keep the identical training setup for both architectures. We grid search for hyper-parameters (lr, hidden_size, and embedder parameterization), even **using more memory** than the CAT baseline, in its global decoder. Even in these simple settings and added advantage, Block Transformer Ho et al. (2024); Yu et al. (2023) fails to solve the task (fig. 3) – instead the model starts to memorize the train points, as seen from train loss and train accuracy – train metrics keep getting better, however, test metrics suffer.

CATs directly pass all the "local" patch/chunk representations directly to the decoder, unlike the block transformer that forces the history to be compressed into fixed dimensional representation. This design choice helps CAT *alleviate the memory bottleneck* that Ho et al. (2024) suffers from where the architecture must compress everything from the past into a single "global" representation to generate the next chunk. Note that this different design choice in CATs does not introduce any memory/compute overhead compared to Block Transformer Ho et al. (2024), it just changes the circuit of the architecture. In fact, CATs don't utilize three different components (embedder, global decoder, local decoder) – it only uses a compressor and a decoder, reducing the design space and (significant) parameter requirements further.

## A.4    RECALL EVALUATION

Here, we evaluate all baselines on all datasets from the EVAPORATE suite of tasks that tests for real-world in-context recall.

| Model | SWDE | FDA | Squad | TriviaQA | Drop | Avg. |
|---|---|---|---|---|---|---|
| Dense | 43.4 | 19.7 | 31.0 | 15.0 | 19.4 | 26.7 |
| Sparse | 20.9 | 6.0 | 20.7 | 15.2 | 19.3 | 16.4 |
| Mamba2 | 13.5 | 4.5 | 24.9 | 13.9 | 17.8 | 14.9 |
| GDN | 18.0 | 6.8 | 25.5 | **15.5** | 17.2 | 16.6 |
| GDN-H1 | 44.0 | 17.8 | **32.9** | 15.4 | **19.8** | 26.0 |
| CAT-4 | **49.1** | **45.1** | 28.3 | 15.0 | 17.9 | **31.1** |
| CAT-8 | 38.2 | 34.8 | 25.9 | 14.0 | 18.3 | 26.2 |
| CAT-16 | 27.5 | 15.4 | 20.4 | 14.8 | 16.9 | 18.9 |
| CAT-32 | 13.2 | 3.2 | 15.8 | 13.0 | 14.3 | 11.9 |

Table 6: Zero-shot performance on real-world in-context recall tasks from EVAPORATE suite, measured upto $4K$ sequence lengths. Note that only SWDE and FDA have long token sequences among the datasets in the suite (others have an average length of $\leq 300$ tokens Arora et al. (2024b)). GDN-Hybrid performs well on short sequences probably due to 2K token long sliding window. In CATs, there is compression even on short sequences.

## A.5    COMPARING CATS WITH PARAMETER MATCHED DENSE TRANSFORMER

Note that Dense 2D is significantly more expensive to run than CAT and is not a fair comparison to CAT or to any other efficient model.

We would again like to reiterate the evaluation criteria in Section 4.1: **raw parameter counts only matter insofar as they affect inference costs. The relevant comparison is what quality a model achieves under a fixed inference cost budget, and not whether two models have the same number of parameters.**

From this inference-cost perspective, Dense 2D is in fact an unfair comparison to CAT (or any other efficient baseline for that matter): it consumes $4\times$ more memory and requires $3\times$ more time to generate than even the least efficient CAT setting (CAT-4). A fair comparison would instead scale down a dense transformer until it is memory-matched to any CAT, say CAT-4 (e.g., by reducing hidden size or depth). Such a dense model would then run under the same memory budget as CAT-4 but would only be worse on real-world recall tasks and language modeling.

Indeed, when we compare our standard dense baseline (Dense D) with CAT, we find that CAT uses $2 - 9\times$ less memory, generates $1.5 - 3\times$ faster, and still matches Dense D on language-modeling evaluations. *This supports our claim that* CAT *can match dense transformers in language modeling while being more efficient.*

**We report Dense 2D performance for transparency and completion of results, rather than as a direct comparison.**

Finally, we emphasize that real-world practice already prioritizes inference cost over parameter counts. Mixture-of-Experts (MoE) models (Agarwal et al., 2025; Yang et al., 2025a) are now widely deployed everywhere despite having $10 - 20\times$ more parameters. When compared at equal total parameter counts, MoEs underperform their dense counterparts (Table 2 in Dai et al. (2024)). Yet they are widely used in production precisely because they deliver better quality at similar or lower inference cost.

CAT should be viewed in this same spirit: what matters is that, for a lower or fixed inference cost budget, CAT achieves competitive or better performance than dense transformers.

| Model | SWDE↑ | FDA↑ | Avg. Recall↑ | LAMBADA↓ | WikiText↓ | FineWeb↓ | Avg. LM Eval↑ |
|---|---|---|---|---|---|---|---|
| Dense D | 43.4 | 19.7 | 32.0 | 38.7 | 19.6 | 17.1 | 42.1 |
| Dense 2D | **53.0** | 34.0 | 44.0 | **35.7** | **16.9** | **15.1** | **45.6** |
| CAT-4 | 49.1 | **45.1** | **47.1** | 38.0 | 18.2 | 16.0 | 43.9 |
| CAT-8 | 38.2 | 34.8 | 36.5 | 37.3 | 18.1 | 15.9 | 44.1 |
| CAT-16 | 27.5 | 15.4 | 21.5 | 36.9 | 18.4 | 16.1 | 44.3 |

Table 7: Evaluation of dense transformer having a $2D = 2048$ hidden size, same as decoder in CAT for completion of results and transparency.

A.6 SPARSE OR SLIDING WINDOW ATTENTION NEEDS MORE LAYERS FOR RECALL

We evaluate models on the synthetic multi-associate query recall (MQAR) task, proposed in Arora et al. (2023a) and further popularized in Arora et al. (2024a). All models use depth of 2 layers, and are trained and tested on sequence lengths upto 256 having varying number of key-value pairs. CAT models use a 1 layer compressor, followed by a 2 layer decoder, with a chunk size of 4, both using model dimension of $D = D_d = 64$ in this case. Note that the state size for CAT is $\frac{N}{C} \cdot D = 4096$ for this particular sequence length and model dimension. Sparse attention uses a chunk size of 4 (for fair comparison with CAT); Sliding window uses a window size of 64.

| Method | Solves? | State Size |
|---|---|---|
| Dense | ✓ | 16384 |
| Sparse | ✗ | 4096 |
| Sliding Window | ✗ | 4096 |
| CAT | ✓ | 4096 |

Table 8: For each method, we report the state size at which the particular method was trained for the MQAR task. Each method was grid searched for best possible hyper-parameters. We use the state size calculations provided in Arora et al. (2024a; 2023a).

In table 8, CAT is able to solve the MQAR task. Notably, we find the sparse attention as well as sliding window attention fail to solve the task at 2 layers, highlighting their dependence on depth.

## A.7 CAT AS A LAYER

To instantiate CAT as a seperate layer in itself, we parameterize the *compressor* as a simple linear projection. We use the dense attention mechanism itself as the *decoder*. Before applying the compression and decoding from compressed chunk representations, we artificially up-project the input embeddings in the layer – this is done following the observation in the main paper that decoding from compressed representations requires higher dimensionality. Please find the actual implementation in the released code. Table 10 reports MQAR accuracy when CAT is used as a layer. We use a fixed chunk size of 4 in this experiment. We use 2 layers of CAT. Rest of the setup follows Arora et al. (2024a).

| Method | Solves? | State Size |
|---|:---:|---:|
| Dense | ✓ | 16384 |
| CAT | ✓ | 4096 |
| CAT (layer) | ✓ | 4096 |

Table 9: CAT instantiated as a seperate layer solves the MQAR task.

## A.8 CAT IS A META SEQUENCE MIXER

CAT solves the MQAR task when the decoder is instantiated as a GDN-Hybrid architecture (1:1 dense-linear ratio) with 2 layers. We use the same setup described in Arora et al. (2024a): using sequences upto 256 with maximum key-values in the sequence. The chunk size used is set to 4.

| Method | Solves? |
|---|:---:|
| Dense | ✓ |
| CAT | ✓ |
| CAT (GDN-HYBRID decoder) | ✓ |

Table 10: The decoder in CAT is replaced with a GDN-HYBRID architecture. The resulting CAT architecture solves the MQAR task.

## A.9 QUALITY AND LATENCY TRADE-OFF

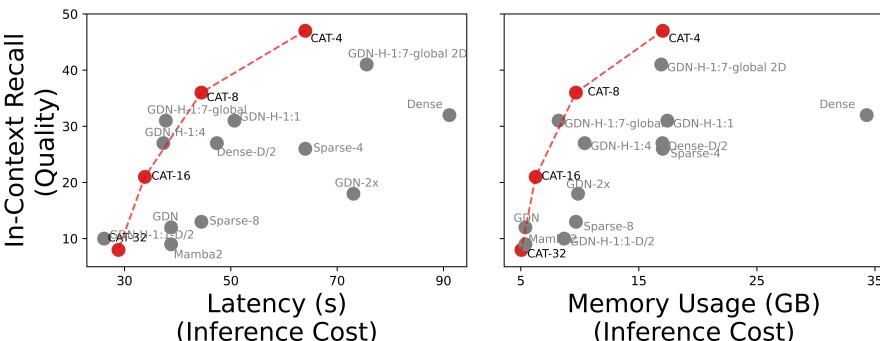

Figure 5: In the main text, we compared different models by measuring what quality (in-context recall) they get for a given inference cost, specifically memory usage, since memory usage is the major inference time bottleneck in increasingly memory-bound GPU workloads (Gholami et al., 2024). We report quality and latency trade-off here for completion. CAT achieves a pareto-frontier in quality (in-context recall) and inference costs (memory usage) trade-off curve across 12 models. At the same time, CAT outperforms most models given a desired latency requirement. CAT achieves this using a *single* model only.

## A.10 CATS SCALE AS WELL AS THEIR DENSE COUNTERPARTS

Figure 6 demonstrates that CATs scale similar to their dense transformer equivalents. We evaluate against three dense transformer scales $\{31M, 92M, 260M\}$, with their CAT equivalents containing parameters $\{95M, 326M, 1B\}$. All models were trained for 15B tokens, under the setup in section 4.

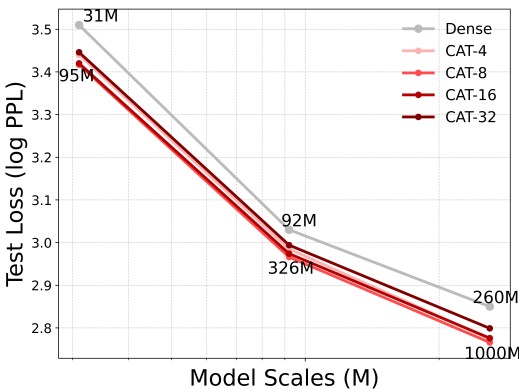

Figure 6: CATs scale like their dense transformer counterparts while being up to $3\times$ faster and $9\times$ more memory-efficient. All CAT curves come from a single model, evaluated at different chunk sizes. **Note that while CAT occupies more parameters, it is still both compute and memory efficient compared to the densee transformer at every scale.**

## A.11 ACROSS CHUNK ANALYSIS

We provide how the validation loss changes within a chunk in Figure 7. We provide averaged results across all chunks. We provide different curves for each chunk size.

Interestingly, across all chunk sizes, the loss is highest when decoding the first token from the compressed representations only. After that token is decoded, the loss decreases steadily as CAT keeps decoding tokens from both compressed representation and raw tokens that appear before inside the chunk.

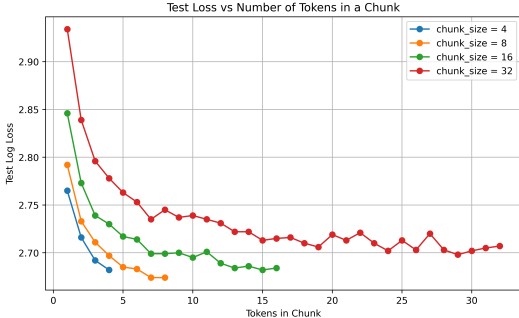

Figure 7: Chunk Analysis

## A.12 S-NIAH-U AND BABILONG RESULTS

Table 11: Accuracy on RULER Hsieh et al. (2024) S-NIAH-U and BabiLong Kuratov et al. (2024) benchmarks.

| Model | S-NIAH-U | | | BabiLong | | | |
|---|---|---|---|---|---|---|---|
| | 1K | 2K | 4K | 0K | 1K | 2K | 4K |
| Dense | **93.6** | 55.7 | 19.8 | **49.0** | 14.0 | 12.0 | 1.0 |
| Sparse | 12.8 | 1.4 | 0.8 | 29.0 | 22.0 | 6.0 | 4.0 |
| Mamba2 | 46.7 | 4.6 | 1.0 | 30.0 | 18.0 | 19.0 | 0.0 |
| GDN | 38.9 | 2.6 | 2.0 | 48.0 | **36.0** | **31.0** | **6.0** |
| GDN-H 1:1 | 50.9 | 5.6 | 2.6 | 35.0 | 10.0 | 2.0 | 1.0 |
| CAT-4 | 79.6 | **59.3** | 46.5 | 46.0 | 22.0 | 9.0 | 1.0 |
| CAT-8 | 68.1 | 57.5 | **47.3** | 46.0 | 19.0 | 9.0 | 5.0 |
| CAT-16 | 10.0 | 6.6 | 3.8 | 31.0 | 5.0 | 8.0 | 5.0 |
| CAT-32 | 0.0 | 0.0 | 0.0 | 17.0 | 10.0 | 7.0 | 5.0 |

## A.13 FINETUNING CATS ON S-NIAH-U

S-NIAH-U is a task where model needs to recall 32 token long UUID strings from the long context. This section reports performance of CATs after task specific finetuning on samples from S-NIAH-U. We only apply the loss on tokens that appear in the answer span. Table 12 reports these results. This is accompanied by loss curves for different CATs depending on chunk size in Figure 8 on this task.

We observe two things: (i) after finetuning, performance goes up significantly for all chunk sizes. This signifies as chunk size increased, compressor in CATs, before finetuning, was not surfacing the *right* information in the chunk representation. (ii) the loss curves during finetuning indicate the same as well, however it still does not go completely to zero, especially for CAT-32. This indicates that there are limits to what information a fixed sized chunk representation can practically learn to surface, justifying its sub-par accuracy on the task.

This problem of not surfacing the *right* information in the chunk representation could be alleviated by more and longer pre-training, or choosing smaller chunk sizes for tasks that require accurate recall.

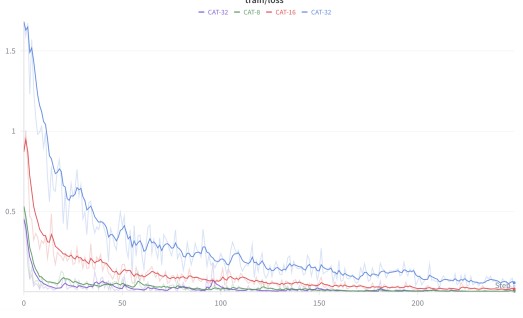

Figure 8: Loss curves when finetuning different CATs on samples from S-NIAH-U task.

| Model | Before | After |
|---|---|---|
| CAT-4 | 46.5 | 97.1 |
| CAT-8 | 47.3 | 97.0 |
| CAT-16 | 3.8 | 94.2 |
| CAT-32 | 0.0 | 64.3 |

Table 12: Performance on 4K sequence length before and after finetuning for different CAT variants.

## B    IMPLEMENTATION DETAILS AND PYTORCH STYLE PSEUDO-CODE

In this section, we discuss some implementation details regarding CATs. We repeat some text presented in the main paper to be self-contained below.

### B.1    TRAINING

**Training:**    While CATs are simple and build on dense transformer abstractions, their naive PyTorch training implementation is very inefficient.

Note that compression of chunks of tokens is efficient since it can be done in parallel, specifically using `torch.vmap(`$f_\theta(\mathbf{c}_i)$`)` for all chunks $\mathbf{c}_i$. This costs a total of $O(\frac{N}{C} \cdot C^2) = O(NC)$ in self-attention compute, which is much better than $O(N^2)$.

But, computing logits for tokens in chunk $\mathbf{c}_i$, that is computing $g_\theta(\mathbf{c}_i \mid f_\theta(\mathbf{c}_1) \cdots f_\theta(\mathbf{c}_{i-1}))$ can be non-trivial since for chunk $\mathbf{c}_i$, we have $i - 1$ past chunk representations $\{f_\theta(\mathbf{c}_1), f_\theta(\mathbf{c}_2) \ldots f_\theta(\mathbf{c}_{i-1})\}$. In other words, there are different number of past chunk representations for every chunk, making shapes variable and as a result, harder to parallelize computation of logits. One could employ a python loop and compute logits for every chunk sequentially, but that would be slow and won't scale. In fact, even if one manages to compute logits for every chunk in parallel, the total self-attention operations in the decoder would be $O(\sum_{i=1}^{\frac{N}{C}}(i + C)^2) = O((\frac{N}{C})^3)$, that is cubic in sequence length. Padding to make shapes constant would make things worse. Thus, naive techniques will not scale.

*With such difficulties in making the training scalable, it may not be surprising that despite the simplicity of CATs, it was not attempted in the community.* Note that unlike CATs, similar architectures Ho et al. (2024); Yu et al. (2023) do not have this problem: computing logits can be naively parallelized due to fixed shapes and self-attention operations scale quadratically due to a single compressed representation for the past.

In CATs, observe that in computing logits chunks $\mathbf{c}_i, \mathbf{c}_{i+1} \ldots \mathbf{c}_{\frac{N}{C}}$, one calculates the same key-values for chunk representations $f_\theta(\mathbf{c}_j)$ in the decoder, where $j < i$. This points to repeated and identical computations. To exploit this observation, we take advantage of a custom attention mask in decoder to calculate logits for all chunks in parallel, and reuse computations done for a past chunk representation to be used for a computations for logits for a future chunk. To be concrete, once we calculate all chunk representations $f_\theta(\mathbf{c}_i)$ in parallel using `torch.vmap`, we insert $f_\theta(\mathbf{c}_i)$s at particular positions in the original sequence: after every chunk $\mathbf{c}_i$, we attach its chunk representation. That is, sequence would look like: $\{\mathbf{c}_1, f_\theta(\mathbf{c}_1), \mathbf{c}_2, f_\theta(\mathbf{c}_2), \ldots \mathbf{c}_i, f_\theta(\mathbf{c}_i) \ldots \}$. Now, we pass this sequence into the decoder during training, with a custom attention mask (see Figure 9) that allows a token in chunk $\mathbf{c}_i$ to attend to previous tokens within that chunk only as well as only to previous chunk representations, which would be $f_\theta(\mathbf{c}_{i-1}), f_\theta(\mathbf{c}_{i-2}) \ldots f_\theta(\mathbf{c}_1)$ only. Any token in chunk $\mathbf{c}_i$ does not attend to raw tokens outside this chunk. This implementation allows re-use of key-values for chunk representations $f_\theta(\mathbf{c}_i)$ for calculation of logits of future chunks, in parallel, making the training of CATs efficient and scalable. We utilize the FlexAttention API Dong et al. (2024) to automatically create a custom kernel for the custom mask (Figure 9). Note that this way of computing logits is quadratic in sequence length but with a constant times better: concretely it is $O(\frac{N}{C} \cdot N + \frac{N}{C} \cdot C^2) = O(\frac{N^2}{C})$, **which is** $C\times$ **better than** $O(N^2)$ (yellow dots in figure 9 provides a visual proof for this cost; number of yellow dots are significantly lower than $\frac{N^2}{2}$). Mathematically the cost of attention in CATs decoder is: $\sum_{i=1}^{N}[\frac{i}{C}] + (i \bmod C) + 1 = O(\frac{N^2}{C})$, where $[.]$ is the floor function, and `mod` is modulo operator.

For a discussion in training throughput, refer to a discussion in Appendix B.5.

```
1
2  def forward(input_ids, targets):
3
4      input_ids = einops.rearrange("b (k c) -> b k c", k=num_chunks, c=
           chunk_size)
5
6      # calculate f(x)
7      # shape of fx: (b, k, D_d)
```

```
8      fx = torch.vmap(f)(input_ids)
9
10     output_logits = list()
11     for i in range(num_chunks): # note that this loop is done in parallel
           with the custom attention mask presented in the appendix
12         # use the previous i+1 fx to predict the current chunk
13         # shape of cur_chunk_logits: (b, 1, l, V)
14         cur_chunk_logits = phi(input_ids[:, i, :], fx[:, :i+1, :])
15         output_logits.append(cur_chunk_logits)
16     output_logits = torch.cat(output_logits, dim=1) # shape: (b, k, c, V)
17     output_logits = einops.rearrange(output_logits, "b k c v -> b (k c) v
           ") # arrange all chunks logits together (or flatten)
18     return torch.nn.functional.cross_entropy(output_logits, targets) #
           return the loss
```

Listing 1: Pseudocode for training step

## B.2 CAT'S TRAINING ATTENTION MASK

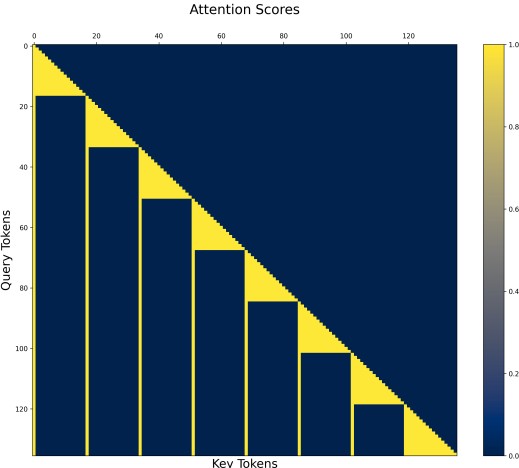

Figure 9: Sequence length is 128, and the chunk size that we use in this particular attention mask is $C = 16$.

Note that attention mask in figure 9 looks very similar to the attention mask as defined in Child et al. (2019), however, in CAT's case: (a) it is not heuristic choice, and (b), tokens in a particular chunk attend to the past $f_\theta(\mathbf{c}_i)$ representations obtained by the compressor, rather than the past token embeddings at that position as done in Child et al. (2019).

## B.3 GENERATION

The decoder during generation attends to atmost $\frac{N}{C} + C$ tokens. Due to compression, CATs can throwaway past chunks of tokens, and only keep their compressed chunk representations in memory. This straightaway results in a big reduction of memory; the KV cache is slashed by a factor of $C$. For even a moderate chunk size of 4, this results in big reductions in memory during generation (Figure 2) compared to a dense transformer. This slash in memory is accompanied by reduced memory accesses a decoder makes in CATs, which is the major bottleneck during generation. Costs for self-attention in CATs decoder scale as $O(\frac{N^2}{C})$, which is again, $C\times$ better than $O(N^2)$ for a dense transformer.

Implementing generation is simpler than training and very similar to how it occurs for a dense transformer. In fact, a pure PyTorch implementation for CATs is on-par with efficient architectures that utilize custom kernels. We inspire our implementation from: https://github.com/meta-pytorch/gpt-fast. Given $i$ chunks of tokens: firstly, torch.vmap over chunks independently to calculate $f_\theta(\mathbf{c}_i)$ in parallel. Then prefill the decoder's KV cache in parallel with the obtained $f_\theta(\mathbf{c}_i)$s. Now generate the next chunk $\mathbf{c}_{i+1}$ autoregressively one token at a time. Note that this uses a simple causal mask since the previous positions are already prefilled with $f_\theta(\mathbf{c}_i)$s, which is required to decode chunk $\mathbf{c}_{i+1}$. Once all the tokens of the chunk $\mathbf{c}_{i+1}$ are generated, calculate $f_\theta(\mathbf{c}_{i+1})$ and prefill the decoder's KV cache just after the position where $f_\theta(\mathbf{c}_i)$ was cached. Now the KV cache is ready for generation of the next chunk $\mathbf{c}_{i+2}$ and this process will continue.

This simple implementation enables CATs to be $1.4 - 3.2\times$ **faster** than the dense transformer while showcasing **upto** $2.2 - 9.5\times$ **lower total memory usage** as one increases chunk sizes.

```python
# https://github.com/pytorch-labs/gpt-fast/blob/7
    dd5661e2adf2edd6a1042a2732dcd3a94064ad8/generate.py#L154
def generate_chunk_by_chunk(
    input_ids
):
    # assume input_ids.shape == (batch_size, 1, chunk_size)

    # declare/reset static KV cache, shape: [batch_size, num_chunks +
        chunk_size, 2, D_d]

    input_pos = 0

    # compress the first chunk (batch_size, 1, chunk_size) -> (batch_size
        , 1, D_d)
    # get fx for the very first chunk
    fx = f(input_ids) # shape of fx: (batch_size, 1, D_d)
    next_token = prefill(fx, input_pos) # prefill at idx 0 with fx in phi

    new_chunks = list()

    for i in range(num_chunks - 1):

        # generate entire chunk using fx that was prefilled earlier in
            phi
        next_chunk = generate_chunk(next_token)
        new_chunks.append(next_chunk.clone())

        # get new fx
        # compress the new obtained chunk
        fx = f(next_chunk) # (batch_size, 1, chunk_size) -> (batch_size,
            1, D_d)

        # prefill again at input_pos
        input_pos += 1
        next_token = prefill(fx, input_pos) # prefill fx at idx `
            input_pos` in phi

    new_chunks = torch.cat(new_chunks)
```

```
34      return new_chunks
```

Listing 2: Pseudocode for generation

### B.4 ADAPTIVE CATS TRAINING DETAILS

To enable training of adaptive CATs, we made some choices that we now describe. In every training iteration, we sample a chunk size uniformly at random and perform loss computation. Further, due to variable size of a chunk in every training iteration, one cannot keep a single projection matrix that projects processed token embeddings in the compressor to a single chunk representation (since shapes for projection matrix would be different for different chunk size). One could tackle this by keeping an independent projection matrix for every chunk size, but we found this didn't work well empirically, possibly due to reduced updates for every chunk size's projection weights (only one chunk size's projection weights are updated per iteration; this is not the case with compressor or the decoder, they are updated every iteration). Instead, we took inspiration from Beyer et al. (2023) where the authors declared a single projection matrix for all chunk sizes, and then linearly interpolated the matrix to the desired shape depending on the current chunk size. This means the linear interpolation is also under `torch.autograd` and is optimized so that the final linearly interpolated projection matrix gives a *good* chunk representation for every chunk size.

### B.5 CAT'S TRAINING THROUGHPUT ANALYSIS

We make use of FlexAttention API to obtain a custom self-attention kernel specifically for the masking scheme section 9. This fused kernel gives a significant boost in training throughput in self-attention costs compared to using a naive PyTorch masked implementation.

That being said, an efficient training kernel can be developed in the future. In our experiments, using FlexAttention did not give significant boosts compared to training speeds using Flash Attention on a dense transformer. This could be due to the fact that speeding up the attention maps (that we use in figure 9) may require different principles than Flash Attention like optimization that Flex Attention might be using under the hood.

Thus, due to the unavailability of an efficient training kernel, theoretical speed ups due to reduction in attention FLOPs in the CAT architecture don't appear in training wall-clock times. Additionally, MLPs in a transformer drive the majority of the FLOPs budget during training at smaller sequence lengths Austin et al. (2025). At a sequence length of 4096, CATs take $\leq 2.35\times$ to train compared to a dense transformer (measured on batch size of 8 with compressor depth of 3, decoder depth of 6, hidden size for compressor $D = 1024$ and hidden size for decoder $D_g = 2D = 2048$ for CAT, compared against dense transformer having depth of 6 and $D = 1024$, on a A100 80 GB PCIe.)

Developing an efficient attention kernel for training CATs is left as future work.

### B.6 TIME TAKEN BY EACH CAT COMPONENT DURING GENERATION

Here we measure time taken by each CAT component during generation, specifically time taken by: decoder attention, decoder FFNs, and time taken by parallel compression. Appendix B.6 provides these results. We use the same setup in benchmarking as described in Section 4. We use a chunk size of 8 for this ablation.

| Component | Time (ms) | Percentage (%) |
|---|---|---|
| Attention in Decoder | 30,817 | 70.1 |
| FFN in Decoder | 11,555 | 26.3 |
| Compression in Compressor | 1,551 | 3.5 |
| Total | 43,932 | 100.0 |

## C    SOME ABLATIONS ON THE CAT ARCHITECTURE

### C.1    ABLATION ON HIDDEN SIZE OF COMPRESSOR

With this ablation, we show that increasing hidden size of the compressor does not help in improving perplexity. We fix $D_g = 1536$ for these experiments. For this ablation, we use a smaller WikiText-103 dataset. Both compressor and decoder use the same depth $L = 6$.

| Chunk Size $C$ | Size of $D_f$ | Perplexity |
|---|---|---|
| 16 | 768 | 17.6 |
|  | 1536 | 17.6 |

Table 13: Comparison of choices of hidden size of compressor on WikiText-103 perplexity.

There is no effect of increasing the hidden size of the compressor. The performance before and after remains the same.

### C.2    ABLATION ON HIDDEN SIZE OF DECODER

We ablate on different choices of $D_g$ along with different chunk sizes in CAT . In this setup, we fix $D_f$ in the compressor, and only vary $D_g$ or $C$ (chunk size). We use WikiText-103 for these experiments. In this setup, $D = 768$. Both compressor and decoder use the same depth of $L = 6$.

| Chunk Size $C$ | Size of $D_g$ | Perplexity |
|---|---|---|
| 4 | $D$ | 19.8 |
|  | $2D$ | 17.4 |
| 8 | $D$ | 20.4 |
|  | $2D$ | 17.7 |
| 16 | $D$ | 20.2 |
|  | $2D$ | 17.6 |

Table 14: Comparison on choices of chunk sizes and sizes of $D_g$ on WikiText-103 perplexity.

We observe that we obtain the best perplexities when we $D_g = 2D$ for the particular chunk size we are using. Using this observation, we used this as our *default* configuration for the FineWeb-Edu experiments.

| Model | $D_f$ | $D_g$ | Perplexity | Avg. recall |
|---|---|---|---|---|
| Dense | $--$ | $D$ | 21.2 | 23.8 |
| CAT | $D$ | $D$ | 23.8 | 13.7 |
| CAT | $D$ | $2D$ | **20.7** | **19.8** |

Table 15: Impact on perplexity and average recall performance of CAT when varying $D_g$. For dense, $D_g$ implies hidden size for itself. Here, $D = 1024$. $D_g = 2D$ gives better perplexity and average recall. We train CAT only at chunk size $C = 8$ for these experiments. All models were trained for 5B tokens with 1K sequence length. Rest of the setup follows Sec. 4.

### C.3    ABLATION ON DEPTH OF THE COMPRESSOR

We ablate on the depth of the compressor. For a fixed chunk-size, $D_f = 768$ (compressor embedding size), $D_g = 1536$ (decoder hidden size), and a fixed depth of the decoder, we vary the compressor depth.

| Chunk Size $C$ | Depth of Compressor | Perplexity |
|---|---|---|
| 8 | 6 | 17.4 |
|   | 3 | 17.4 |
| 16 | 6 | 17.8 |
|    | 3 | 17.7 |

Table 16: Comparison on choices of depth of the compressor across different chunk sizes $C$ on WikiText-103.

We have an interesting observation that one can reduce the depth of the compressor without sacrificing on the downstream perplexity. This could mean one can compress small chunks of tokens without a requiring high capacity. In our generation benchmarks, we observed that compressor depth play less of a role in latency as compared to the decoder depth (since we compress tokens in parallel using one transformer call). That being said, compressor depth does play a significant role in training costs (due to the MLP training costs in the compressor). Therefore, reducing compressor depth goes into overall advantage for the CAT architecture.

However, what is the limit, and can one go to even a 1 layer of compressor is an interesting question to ask. There might be some lower bound on the compressor depth to start compressing chunks of tokens, but we leave this to future work.

# D    MORE EXPERIMENT DETAILS

Here we provide more details about the experiments done in the main text.

## D.1    BASELINES

| Model | Total (M) | Embedding (M) | Non-Embedding (M) |
|---|---|---|---|
| Dense | 260 | 50 | 210 |
| Dense-D/2 | 92 | 25 | 70 |
| Mamba2 | 260 | 50 | 210 |
| GDN | 310 | 50 | 260 |
| GDN-2x | 310 | 50 | 260 |
| GDN-Hybrid 1:1 | 280 | 50 | 230 |
| GDN-Hybrid 1:1 D/2 | 111 | 25 | 86 |
| GDN-Hybrid 1:4 | 280 | 50 | 230 |
| GDN-Hybrid global 1:7 | 300 | 50 | 250 |
| GDN-Hybrid global 1:7 2D | 820 | 100 | 720 |
| Sparse-4 | 820 | 100 | 720 |
| Sparse-8 | 820 | 100 | 720 |
| CAT-4/8/16/32 | 150 + 820 | 50 + 100 | 100 + 720 |

Table 17: Model parameter sizes in millions, separated into embedding and non-embedding parameters. Parameters for CATs consists of cost of compressor + cost of decoder.

1. Dense transformer (or Transformer++) Vaswani et al. (2017); Touvron et al. (2023): We use rotary position embeddings along with the FlashAttention kernel to perform self-attention. The MLP is a SwiGLU MLP Touvron et al. (2023).

2. Sparse transformer Child et al. (2019): Follows the Dense transformer configuration, except the attention mask used. Moreover, we used $D = 2 \cdot 1024 = 2048$ for this baseline for a fair comparison with CATs. We used FlexAttention API to create optimized Flash Attention like kernel for this. We use a stride length of 8 that tries to compete with CAT-8.

3. MAMBA2 Dao & Gu (2024): The model uses 2 Mamba mixer per layer. All layers use the MAMBA2 block without any mixing any attention. The `expand` is set to 2, $d_{state} = 128$, and convolution $k = 4$. Activations used are `SiLU`. We use the official codebase for MAMBA2 generation throughput and memory benchmarking: `https://github.com/state-spaces/mamba` and code from: `https://github.com/fla-org/flash-linear-attention` for training.

4. Gated Delta Net Yang et al. (2025b): We use the implementation provided at `https://github.com/fla-org/flash-linear-attention` for training. We use `head_dim` as 128 and `num_heads` as 8 (same as MAMBA2 above). For the hybrid version (GDN-HYBRID 1:1), we use sliding window layers at every other layer with a sliding window size of 2048. GDN-2X uses a `head_dim` as 256 and `num_heads` as 4. GDN-HYBRID 1:7 GLOBAL uses linear-dense attention ratio as 1:7 with global attention, and GDN-HYBRID 1:7 2D GLOBAL uses 2× the hidden size with global attention. GDN-HYBRID 1:4 uses 1:4 ratio with sliding window of 2K. Finally, GDN-HYBRID 1:1 D/2 uses 2× less model dimension.

## D.2    DATASETS

Following common practices done in Gu & Dao (2023); Dao & Gu (2024); Arora et al. (2024a); Yang et al. (2025b), we evaluate all models on multiple common sense reasoning benchmarks: PIQA Bisk et al. (2020), HellaSwag Zellers et al. (2019), ARC-challenge Clark et al. (2018), WinoGrande Sakaguchi et al. (2021) and measure perplexity on WikiText-103 Merity et al. (2016)and LAM-BADA Paperno et al. (2016). In Table 5, HS denotes HellaSwag, PQ denotes PIQA, AE denotes

ARC-Easy, AC denotes ARC-Challenge, WG denotes Winogrande, OQA denotes OpenBookQA, LMB denotes LAMBADA, Wiki denotes WikiText, and FW denotes FineWeb-Edu.

We evaluate on tasks from LongBench Bai et al. (2023) where each abbrevation in table 3 stands for: QAS: qasper, MQA: multifieldqa_en, HQA: hotpotqa, 2WMQ: 2wikimqa, TQA: triviaqa, TREC: trec split of LongBench.

To measure real-world recall accuracy, we use datasets used in Arora et al. (2024a;b). Namely these consists of SWDE Lockard et al. (2019) for structured HTML relation extraction and several question answering datasets including SQuAD Rajpurkar et al. (2018), TriviQA Joshi et al. (2017), DROP Dua et al. (2019) and FDA Arora et al. (2023b). Since our pretrained models are small, we use the Cloze Completion Formatting prompts provided by Arora et al. (2024b).

We evaluate on tasks from the needle-in-haystack benchmark RULER Hsieh et al. (2024).

Additionally, we evaluate on datasets from the LongBench benchmark Bai et al. (2023) to evaluate long-context understanding.

Finally, to evaluate baselines on state-tracking tasks, we used the BabiLong benchmark Kuratov et al. (2024). Due to relatively small scale of our setup, we were only able to evaluate on qa1 subset, since for other complex subsets, all baselines failed.

### D.3 GENERATION

Both dense transformer and CAT use FlexAttention API causal dot product kernels. We use the script provided in Dao & Gu (2024) to benchmark[6] Mamba2, GatedDeltaNet and GatedDeltaNet-Hybrid. All benchmarks used a prefill of $8$ tokens. All benchmarks were run using a single NVIDIA A100 80GB PCIe, and use CUDA cache graphs for the next-token prediction.

### D.4 MAIN FIGURE DETAILS

Figures 1a and 5 reports memory usage at 2K sequence length since both SWDE and FDA datasets have queries with context length upto 2K. The latencies in Figure 5 are reported at maximum sequence length of 4K.

---

[6]github.com/state-spaces/mamba

## E   EXTENDED RELATED WORK

**Reducing self-attention costs:**   Reducing the cost of self-attention enables scaling transformers to large contexts and has been the focus of much work Child et al. (2019); Parmar et al. (2018); Beltagy et al. (2020); Jiang et al. (2023). Common techniques include *heuristically* defined sparse attention maps Child et al. (2019); Zaheer et al. (2020) or a sliding window Jiang et al. (2023) in order to reduce the tokens being attended to. The compute required (and in some cases, memory) for attention go down, however, compromising with the expressivity of the model. In turn, to achieve performance similar to that of full-attention, efficient models either require big window sizes (making their memory costs large again) (Arora et al., 2024a) or more layers (in case of sparse or sliding window attention, see App. A.6 and Tab. 2).

Shazeer (2019) proposes use of single or reduced key and value heads in the self-attention block, more commonly known as Grouped Query Attention (only one key/value head) or Multi Query Attention (reduced key/value heads). This results in reduction of memory with seemingly no loss in downstream performance, making this a popular choice in latest model releases Yang et al. (2025a). That being said, one could use the same technique inside CAT's decoder (and compressor) self-attention block, making it complimentary.

Concurrent works like Yuan et al. (2025) reduce attention compute by attending to compressed past tokens as well as to specific blocks of uncompressed tokens in the past. This is similar in spirit to our work, however, in the case of Yuan et al. (2025), there are no memory savings during inference.

Some works Rae et al. (2020); Chevalier et al. (2023) explored recurrent formulations of a transformer to enable processing of longer sequences on limited compute by compressing past context. However, training sequence models in a recurrent fashion has its own challenges, back-propagation through time (BPTT) being the most important one. More recently Geiping et al. (2025) had to use very careful weight initialization, truncated gradients, small learning rates and careful placement and tuning of norms to train a large-scale recurrent architecture in a stable manner and prevent optimization collapse. Nevertheless, these techniques are complementary to CAT.

Alternatively, one can optimize the computation of full-attention to directly reduce wall-clock time and memory by leveraging hardware advancements. For example, Dao et al. (2022) compute attention in blockwise manner and exploit the nature of online softmax Milakov & Gimelshein (2018) which removes the need to instantiate the entire $QK^T$ matrix and reduce calls to slow-read part of the GPU memory. As we utilize the attention mechanism as is, any reductions in cost due to hardware optimization that apply to the attention mechanism also proportionally reduce the cost of CAT models.

Finally, plethora of works have tackled reducing compute and memory requirements of a transformer in a *post-hoc* manner i.e. after it has been trained using full-attention (also called *training-free* sparse attention Nawrot et al. (2025)). Common techniques include prefill-time sparsification (vertical/slash/block; adaptive) and decode-time KV-cache selection/eviction (e.g. Li et al. (2024b); Tang et al. (2024)). However, because models are trained dense but run sparse, train–test mismatch can hurt downstream performance. Still, these works are orthogonal to CAT and can be layered on CAT's decoder, making them complementary.

**Linear attention and state-space models:**   A different line of work reduces the generation cost of transformers by limiting the recurrent state, which is the vector required to decode each token. Self-attention keeps track of the entire context (or the KV cache) meaning that the recurrent state increases in size with each decoded token. Works like Arora et al. (2024a); Katharopoulos et al. (2020) linearize attention to make a fixed-size recurrent state that can be updated via simple averaging; the technique is to approximate self-attention with linear operations of query, key, and value vectors transformed through a feature map. The choice of the feature map falls to the user and approximating attention well requires the feature map to be large in size, which can counteract the gains in computational costs achieved by the linearization.

Alternatively, one can replace attention with linear or pseudo-linear sequence mixers such as state-space models (SSMs) Gu et al. (2021); Sun et al. (2023), gated convolutions Fu et al. (2022); Poli et al. (2023) and input-dependent recurrent Peng et al. (2023); Gu & Dao (2023) and more recently Yang et al. (2025b).

Typical implementations of linear attention and state-space models do achieve impressive reductions in generation costs and memory, but restrict the expressivity to the extent that these models do not solve in-context recall tasks without large recurrent state sizes Arora et al. (2024a; 2023a), or without composing with other sequence mixers, such as local sliding window attention (Arora et al., 2024a; Yang et al., 2025b). Choosing such a composition again falls back to the user, complicating the design process. Additionally, this process trades-off computation costs for performance because the attention layers that improve recall performance also come with larger time and memory costs.

Unlike the works discussed above, CATs require no complicated changes to the attention mechanism itself. CATs rely on the fact that natural language is redundant and can be compressed Zipf (2016); Shannon (1951). Instead of relying manual approximations of history or utilizing any heuristic choice for feature maps, we let the model and optimization decide what the history should be using learned compression. Moreover, its unclear how much memory and compute a downstream task requires, making the adaptive property of CATs much desirable, which no other baselines provide.

**Hierarchical transformers:** Many previous works Pappagari et al. (2019); Han et al. (2021); Dai et al. (2020) have explored employing hierarchy in transformers for creating representations representations for documents/images, where a *local* encoder transformer processed parts of the document/image independently. Later works Nawrot et al. (2021; 2022); Slagle (2024) explored downsample-then-upsample approach (*hour-glass* like structure), where the sequence is downsampled into *coarse* tokens followed by upsampling into *fine-grained* tokens before being decoded. Due to the *hour-glass* structure, there are compute savings during training, but during generation, the architecture must maintain a cache for all the past tokens, leading to significant memory accesses. Concurrently, Hwang et al. (2025) explored a dynamic and end-to-end learned strategy for chunking in *hour-glass* like architectures.

Different from above, works like Ho et al. (2024); Yu et al. (2023) break up the modeling of a sequence into chunks/patches, where each chunk is modeled independently of each other given the previous "global" chunk embedding. An embedder first compresses each chunk independently, then these "local" chunk embeddings are passed to a "global" model where each "local" chunk embedding attends to past "local" chunk embeddings, forming a "global" chunk embedding. Each "global" chunk embedding is then passed to a decoder that is responsible for generating the next chunk.

On first glance, CATs might appear similar to above works, specifically Ho et al. (2024); Yu et al. (2023), however the subtle but salient difference is: one directly feeds **all the previous "local" chunk/patch representations** directly to the decoder in CAT, whereas in works like Ho et al. (2024), one feeds in just the previous "global" chunk representation outputted by a "global" model to the decoder. This architectural choice of passing *all* the compressed local chunks from the past directly to the decoder allows CATs to solve long-range recall tasks with ease while maintaining efficiency, whereas Ho et al. (2024) is plagued by *learnability* problems (even in toy recall tasks) due to constant size compression of history. Additionally, CATs don't utilize three different components (embedder, global decoder, local decoder) – it only uses a compressor and a decoder, reducing the design space and (significant) parameter requirements further.

Additionally, Yen (2024) extend the cache by using a modified encoder-decoder architecture, where decoder attends directly to final activations of a smaller fixed encoder, without any compression.

Finally, Barrault et al. (2024) suggest learning "concepts" instead of tokens by modeling the latent representation of language produced by pushing the token sequence through a large sentence embedder. The focus of this work is to decouple the modeling of the low-level details in each language, like tense and grammar, from the larger concept space that is shared across languages. In contrast, the goal with CAT is to reduce the cost of modeling sequences and can be used as a plug-and-play replacement to the latent concept model. Moreover, the encoder in Barrault et al. (2024) is an auto-encoder, that might keep irrelevant information in the chunk representation. Compressor in CATs only keeps information that is predictive of the future chunks.

**Adaptive architectures:** Kusupati et al. (2022); Devvrit et al. (2023) learns representations during training time that can work at different granularity during test-time, yielding adaptivity to the learned architecture. However, coarser granularity of *Matryoshka* representations result in loss of language modeling performance (in terms of perplexity) Devvrit et al. (2023). That being said, one could apply similar approaches to CATs making them complimentary. CATs use the same high-level

approach described in Beyer et al. (2023): learn a single model that can work for various patch sizes at once depending on the downstream use-case at test-time. However, Beyer et al. (2023) worked with image classification tasks; CATs deal with language modeling and generation.

