# OpenReview forum: "Attention and Compression is all you need for Controllably Efficient Language Models"
_ICLR.cc/2026/Conference — Submitted to ICLR 2026_

### Official Review · Reviewer_Gs4J · 2025-10-31

**Soundness:** 3
**Presentation:** 3
**Contribution:** 2
**Rating:** 4
**Confidence:** 4

**Summary:**

The paper proposes Compress & Attend Transformers (CATs), an adaptive architecture that decodes each chunk of tokens while attending to compressed representations of all previous chunks. Training interleaves compressed chunk tokens with raw tokens and applies a custom attention mask so each token looks only within‑chunk plus compressed past, enabling parallel compression and reducing decoder attention cost. On a single model trained with multiple chunk sizes, the authors show test‑time control of the quality/efficiency trade‑off via a special indicator token. Empirically, with a 12‑layer decoder widened to 2× hidden size and a small 3‑layer compressor, CATs match or beat several efficient baselines on long‑context recall and some language‑understanding tasks.

**Strengths:**

1. Simple, adaptive mechanism with a clear knob: CATs realize a neat dial‑a‑budget via chunk size, using a learnable indicator token plus a shared marker to separate compressed from raw tokens.
2. Scalable training trick that avoids recurrence: The paper’s interleaving of compressed vectors into the sequence with a custom mask reuses K/V for chunk representations and reduces decoder attention.
3. Ablations and design transparency. Useful findings include: decoder width matters (D -> 2D helps LM perplexity), compressor depth has minor impact, and large chunks trade recall for speed.

**Weaknesses:**

1. Main CAT model is 1B params (wider decoder + compressor) vs 260–310M for dense/linear baselines. While serving memory is KV‑dominated, LM perplexity gains could partially reflect capacity. Indeed, a parameter‑matched dense (2D) outperforms CAT on several LM metrics, weakening the claim that CATs match dense on LM while being faster. Please provide more apples‑to‑apples comparisons at equal params/training FLOPs.
2. Cross‑chunk fidelity trade‑offs not fully characterized. By design the decoder never attends to raw tokens outside the current chunk. Please add ablations on cross‑chunk phenomena.
3. Scope limited to 4K contexts. Most results cap at 4K because of training context. Since the motivation is long‑context efficiency/recall, evaluations at 8–32K (even with extrapolation) would strengthen claims.

**Questions:**

N/A

---

> ### Author Response · Authors · 2025-11-25
>
> ### Weakness 1
>
> > Main CAT model is 1B params (wider decoder + compressor) vs 260–310M for dense/linear baselines ...
>
> and
>
> > Please provide more apples‑to‑apples comparisons at equal params/training FLOPs ...
>
> We would like to direct the reviewer to our general response 1 [here](https://openreview.net/forum?id=6rYa2BUnTt&noteId=3r23OwI7TD), where we make an _apples-to-apples comparison between different architectures_. Specifically, we note that **raw parameter counts only matter so much as their effect on inference cost, and the right comparison is _what_ quality a model obtains for a given inference cost.**
>
> Hence, we compare different models by _matching inference costs_. Further, for a more exhaustive evaluation and to prevent any cherry picking, we pre-trained additional models for each model type by changing their model configurations (scaling up their recurrent state size, or model dimension, or changing their hybrid ratios). In total, this results in around 12 models that we compare CAT against. These models have a range of parameter counts, from ~300M to upto ~800M, with varying inference costs. Every model type has atleast ≥2 model configurations to ensure fair and broader evaluation. These models are different points on the quality-inference costs trade-off curve.
>
> We observe that **CAT obtains a pareto frontier in the quality-inference costs trade-off curve across all models, including the scaled up ones. Importantly this is achieved using just a single model.**
>
> In fact, under matched inference costs, **CAT outperforms models like `GDN-H-1:7-global-2D` and `Sparse-4/8` that use similar parameters as CAT.** Note that these, including others are different models that occupy different locations on the quality-inference cost curve. Any prior baseline requires retraining from scratch to target different inference budgets, whereas **CAT makes it easy to _traverse_ this trade-off directly at test-time.**
>
>
> > While serving memory is KV‑dominated, LM perplexity gains could partially reflect capacity. Indeed, a parameter‑matched dense (2D) outperforms CAT on several LM metrics, weakening the claim that CATs match dense on LM while being faster.
>
> We would like to direct the reviewer to our second general response 2 ([here](https://openreview.net/forum?id=6rYa2BUnTt&noteId=WgvmGO1piP)), where we discuss exactly this concern.
>
> Further, we agree with the reviewer that LM perplexity gains be due to capacity. However, we would like to mention that it is the architecture of CAT that allows for this additional capacity without increasing inference costs, and should be considered as a _feature_ that CAT provides (similar in vein to MoEs).

---

> > ### Author Response · Authors · 2025-11-25
> >
> > ### Weakness 2
> >
> > > Cross‑chunk fidelity trade‑offs not fully characterized. By design the decoder never attends to raw tokens outside the current chunk. Please add ablations on cross‑chunk phenomena.
> >
> > The reviewer is correct that CAT's decoder never attends to raw tokens outside the current chunk.
> >
> > We provide how the validation loss changes within a chunk. [Here](https://ibb.co/zTN6Mx37) is a link for the figure. We provide averaged results across all chunks. We provide different curves for each chunk size.
> > The reviewer is correct when they say CAT’s decoder never attends to raw tokens outside the current chunk.
> >
> > Interestingly, across all chunk sizes, the loss is highest when decoding the first token from the compressed representations only.
> > After that token is decoded, the loss decreases steadily as CAT keeps decoding tokens from both compressed representation and raw tokens that appear before inside the chunk.
> >
> > We will add this analysis in our final manuscript.

---

> > > ### Author Response · Authors · 2025-11-25
> > >
> > > ### Weakness 3
> > >
> > > We provide scaled up results on 8K sequence length.
> > > All models were trained from scratch on 8K sequence length for 30B tokens on FineWeb-Edu.
> > > We additionally increased the number of layers to 18 for all models.
> > >
> > >
> > > | Model  | NIAH-N 2K   | NIAH-N 4K   | NIAH-N 8K   | Avg. LM evals |
> > > |--------|------|------|------|---------|
> > > | Dense  | 99.3 | 59.5 | 37.9 | 44      |
> > > | GDN-Hybrid 1:1 | **99.7** | 53.2 | 24.9 | 44.4    |
> > > | CAT-4  | 99   | **90.4** | **98**   | 44.6    |
> > > | CAT-8  | 94.4 | 92   | 87   | 44.5    |
> > > | CAT-16 | 82.7 | 82.7 | 57.8 | 45.3    |
> > > | CAT-32 | 71.8 | 42.9 | 26.2 | **45.8**    |
> > >
> > > CAT-4 outperforms Dense, whereas CAT-8 outperforms GDN-Hybrid especially at a longer context of 8K.  Note that all CATs reported above are a single model.
> > >
> > > We will add these results in our final manuscript.

---

### Official Review · Reviewer_McFn · 2025-10-31

**Soundness:** 2
**Presentation:** 3
**Contribution:** 3
**Rating:** 2
**Confidence:** 3

**Summary:**

This paper proposes the Compress & Attend Transformer (CAT) as an efficient alternative to standard Transformers. CAT first partitions the input tokens into chunks and employs a lightweight compressor (bid-Transformer + projection) to obtain a compressed representation for each chunk. In the main decoder LLM, each token attends to both the local tokens within chunk and the compressed representations of all preceding chunks. This design leads to computational and memory savings proportional to the chunk size during inference. Furthermore, by training CAT with multiple chunk sizes, the resulting adaptive CAT can flexibly operate under different budgets at test time by adjusting the chunk size. Experimental results demonstrate that CAT achieves stronger language modeling and in-context recall performance compared to other efficient baselines, while delivering significant inference efficiency gains over dense Transformers.

**Strengths:**

1. The paper provides a clear and coherent exposition of the proposed CAT methodology. The compress-and-decode design is both simple and effective, offering distinctive advantages over prior approaches such as sparse attention and fixed-size recursive compression.

2. The training strategy of CAT is relatively novel. Compared to dense Transformers, it achieves training efficiency gains while maintaining parallelism. The use of multiple chunk sizes during training presents a well-motivated approach to adaptive compute allocation at test time.

3. The evaluation is comprehensive, covering language modeling, synthetic and real-world recall, and long-context QA, etc.

**Weaknesses:**

1. **Unfair comparison:**
To offset performance gaps, the authors doubled the decoder dimension for CAT, resulting in **non-proportional parameter counts** across models. CAT is both **deeper and wider**, which inherently benefits in-context and knowledge capability, especially for smaller models (e.g., commonsense reasoning sensitive to these factors). In the more balanced comparison of **Table 7**, CAT actually shows some disadvantage. While the authors argue that larger CAT models remain more efficient than Transformers, this makes comparisons with other efficient baselines **meaningless**—those baselines could similarly scale up parameters while maintaining efficiency gains.

2. **Weak and incomplete baselines:**
The efficiency analysis (Fig. 3) only reports results up to 4K length, where CAT’s **quadratic complexity** is not yet dominant. For longer contexts, CAT would exhibit significant efficiency degradation compared to baselines such as GDN or GDN-H1. Thus, improvements in recall primarily indicate a **quality–efficiency trade-off** rather than a strict superiority. Moreover, modern practice no longer relies on fixed-size state models alone, and **hybrid linear + global attention** architectures are now the norm. A fairer comparison should involve a **GDN-Hybrid with dense attention** (e.g., 1:(C-1) hybrid ratios), possibly with 2x model dimensions , which would achieve similar compute savings (wrt. CAT-C) at normal sequence lengths.

3. **Limited efficiency gains:**
Reported improvements mainly arise from CAT-4 and CAT-8, which yield roughly ½× and ¼× compute/memory reductions, respectively. However, layerwise-hybrid efficient baselines (w/ swa or linear attention) may offer better trade-offs. Such layerwise scheme could extend the efficiency advantage much further (e.g., up to 1:7 or beyond), suggesting that CAT’s practical efficiency benefits might be moderate at best.

**Questions:**

1. CAT partitions the input into chunks of fixed sizes. Could this design potentially limit or even harm model performance, considering that tokens within a chunk may be totally semantically unrelated? Would a more input-aware or adaptive chunking strategy, as explored in [1], be a more reasonable choice?

2. Does this coarse-grained, chunk-level compression inherently disadvantage tasks that require fine-grained token-level reasoning or recall? CAT appears conceptually similar to NSA [2] without its sparse-attention branch. However, that branch which handles fine-grained long-range interactions, has proven crucial for long-context or information-dense tasks.

3. The paper does not specify the exact configuration of the "sparse attention" baseline. If it only uses a static, input-agnostic sparse pattern, the comparison may be unfair, since CAT’s compression involves contextual interaction (via a bidirectional Transformer). For language modeling tasks (that is considered to be semantically rich), dynamic sparse attention methods (e.g., [3]) are generally more representative baselines than static patterns.

---

[1]  Dynamic chunking for end-to-end hierarchical sequence modeling, 2025.
[2] Native Sparse Attention: Hardware-Aligned and Natively Trainable Sparse Attention, 2025.
[3] H2O: Heavy-Hitter Oracle for Efficient Generative Inference of Large Language Models, 2023.

---

> ### Author Response · Authors · 2025-11-25
>
> ### Weakness 1
> > Unfair comparison: To offset performance gaps, the authors doubled the decoder dimension for CAT, resulting in non-proportional parameter counts across models ...
>
> and
>
> > While the authors argue that larger CAT models remain more efficient than Transformers, this makes comparisons with other efficient baselines meaningless—those baselines could similarly scale up parameters while maintaining efficiency gains.
>
> We thank the reviewer for raising the concern about unfair comparisons with the baselines. We would like to direct the reviewer to our general response 1 ([here](https://openreview.net/forum?id=6rYa2BUnTt&noteId=3r23OwI7TD)) addressing exactly this concern. In short, we discuss how **raw parameter counts only matter so much as their effect on inference cost, and the right comparison is what quality one obtains for a given inference cost.** Hence, we compare different baselines by _matching inference costs_.
>
> Further, we agree with the reviewer that other baselines could similarly scale up while maintaining efficiency. As a result, we pre-train seven additional models from scratch by changing model configurations of each of the model types.
>
> To be specific, we pre-trained the following new models from scratch for each model type:
>
> - GDN-Hybrid
>     - `GDN-H-1:7-global`: Change hybrid ratios of GDN-Hybrid to 1:7 (i.e. `1:(C-1)` where `C=8`) with global attention (suggested by the reviewer)
>     - `GDN-H-1:7-global-2D`: Change hybrid ratios of GDN-Hybrid to 1:7 with global attention, and increase the model dimension by 2x. **Note that this model has a similar number of parameters as CAT**. (suggested by the reviewer)
>    - `GDN-H-1:4`: Change hybrid ratios of GDN-Hybrid to 1:4
>
>     - `GDN-H-1:1-D/2`: Decrease the model dimension by 2x.
> - GDN
>     - `GDN-2x`: Increase the recurrent state size by 2x
> - Sparse
>     - `Sparse-4`: Decrease the sparsity by 2x. **This model has a similar number of parameters as CAT**.
> - Dense
>     - `Dense-D/2`: Decrease the model dimension by 2x
>
>
>
> This results in a total of 12 models that we comapre CAT have varying inference costs.
>
> We find that a single CAT model either matches or outperforms all these models in quality (in-context recall), when compared under the same inference cost (memory usage) achieving a pareto-frontier in quality-inference costs trade-off (see general response 1 [here](https://openreview.net/forum?id=6rYa2BUnTt&noteId=3r23OwI7TD)).
>
> Further, **CAT outperforms the "parameter matched" `GDN-H-1:7-global-2D`, `Sparse-4`, `Sparse-8` when compared under matched inference costs.**
>
> _Importantly, any prior baseline requires re-training from scratch to target these different inference budgets, however CAT provides a simple knob at test-time for this._

---

> > ### Author Response · Authors · 2025-11-25
> >
> > ### Weakness 1 continued
> > > CAT is both deeper and wider, which inherently benefits in-context and knowledge capability, especially for smaller models (e.g., commonsense reasoning sensitive to these factors).
> >
> > We agree with the reviewer that CAT is wider, and can be thought of as being _deeper_. We would like to mention that it is the architecture of CAT that allows for a wider decoder and an additional compressor without increasing inference costs, and should be considered as a _feature_ that CAT provides.
> >
> >
> > > In the more balanced comparison of Table 7, CAT actually shows some disadvantage.
> >
> > We would like to direct the reviewer to our general response 2 ([here](https://openreview.net/forum?id=6rYa2BUnTt&noteId=WgvmGO1piP)), where we discuss exactly this concern.

---

> ### Author Response · Authors · 2025-11-25
> **Weakness 2**
>
> ### Weakness 2
>
> > The efficiency analysis (Fig. 3) only reports results up to 4K length, where CAT’s quadratic complexity is not yet dominant. For longer contexts, CAT would exhibit significant efficiency degradation compared to baselines such as GDN or GDN-H1. Thus, improvements in recall primarily indicate a quality–efficiency trade-off rather than a strict superiority.
>
> While we agree that CAT is still quadratic, using a larger chunk size at longer sequence lengths can alleviate this **significantly** by an order of magnitude (similar to figure 3 where increasing chunk size gives significant speed ups in generation). Further, the adaptive nature of CAT makes it a very easy switch, without any retraining.
>
> Further, while accuracy can drop at longer context length when using a large chunk CAT, we note that the same problem exists for linear attention or any fixed memory architecture (although CAT shows a much more graceful drop in quality). This means that to process sequences at longer contexts, one perhaps requires fundamentally more compute and memory indicating a fundamental trade-off [1] as the reviewer pointed out. _CAT just makes it easy to traverse this trade-off at test-time._
>
>
> [1] Arora, Simran, et al. "Simple linear attention language models balance the recall-throughput tradeoff." arXiv preprint arXiv:2402.18668 (2024).
>
> > Moreover, modern practice no longer relies on fixed-size state models alone, and hybrid linear + global attention architectures are now the norm. A fairer comparison should involve a GDN-Hybrid with dense attention (e.g., 1:(C-1) hybrid ratios), possibly with 2x model dimensions , which would achieve similar compute savings (wrt. CAT-C) at normal sequence lengths.
>
> We thank the reviewer for suggesting these. We would like to direct the reviewer to our general response 1 [here](https://openreview.net/forum?id=6rYa2BUnTt&noteId=3r23OwI7TD) and our response to Weakness 1 where we pre-train scaled up versions of the model types as suggested by the reviewer.
>
> **Under matched inference costs, CAT outperforms almost every model, all trained with different model configurations targetting different the inference budgets. Importantly, CAT achieves this with just a single model only.**

---

> > ### Author Response · Authors · 2025-11-25
> >
> > ### Weakness 3
> >
> > > Reported improvements mainly arise from CAT-4 and CAT-8, which yield roughly ½× and ¼× compute/memory reductions
> >
> > While CAT-4 is best in absolute sense, as shown in our general response 1 [here](https://openreview.net/forum?id=6rYa2BUnTt&noteId=3r23OwI7TD), CAT matches or shows improvements in quality (in-context recall) relative to all baselines under varying inference budgets. Hence, we respectfully disagree that improvements only arise from CAT-4 or CAT-8.
> >
> > > However, layerwise-hybrid efficient baselines (w/ swa or linear attention) may offer better trade-offs. Such layerwise scheme could extend the efficiency advantage much further (e.g., up to 1:7 or beyond), suggesting that CAT’s practical efficiency benefits might be moderate at best.
> >
> > We have included these respective models (and more) in our general response 1 [here](https://openreview.net/forum?id=6rYa2BUnTt&noteId=3r23OwI7TD).
> > While these models occur at different locations on the quality-inference cost trade-off curve, they still match or fall below the CAT’s pareto frontier.
> >
> > **More importantly, this need to carefully tune hybrid ratios (1:1, 1:7 or any other 1:X etc.) for a desired quality-efficiency trade-off is precisely what CAT is designed to avoid at scale: CAT exposes multiple trade-offs directly at test time without retraining.**
> >
> > Further, as shown in general response 1 [here](https://openreview.net/forum?id=6rYa2BUnTt&noteId=3r23OwI7TD), CAT can be thought of as a **meta sequence mixer** i.e. general arrangement of compressor and decoder blocks. This compressor and decoder themselves could be made up of any architecture, meaning all the efficiency advantages of different architectures due to "layerwise schemes" (e.g, of GDN-Hybrid) can be directly ported to CAT. This makes CAT complementary to existing efficient architectures and all the excellent research being done on efficient architectures can be readily used in CAT.
> >
> > Additionally in the general response 1 [here](https://openreview.net/forum?id=6rYa2BUnTt&noteId=3r23OwI7TD), we discuss how CAT can be instantiated as a layer, which can be used as a drop-in replacement layer in any architecture. This unlocks the same interesting possibilities of new hybrid architectures of using dense attention alongside CAT attention in a single architecture; the same benefits that linear attention enjoys. One can use the same "layerwise schemes" and even mix all three layers (dense attention, linear attention and CAT attention) in a single architecture to have benefits of all.

---

> ### Author Response · Authors · 2025-11-25
>
> ### Questions
>
> > 1. CAT partitions the input into chunks of fixed sizes. Could this design potentially limit or even harm model performance, considering that tokens within a chunk may be totally semantically unrelated? Would a more input-aware or adaptive chunking strategy, as explored in [1], be a more reasonable choice?
>
> We agree with the reviewer that a dynamic strategy to decide chunk sizes based on inputs could potentially improve quality. In our work, we used fixed chunk sizes to simplify the design process of CAT, and found it to work well. This can be an orthogonal frutiful future research to pursue.
>
> > 2. Does this coarse-grained, chunk-level compression inherently disadvantage tasks that require fine-grained token-level reasoning or recall?
>
> While one may think CAT would lose fine-grained token-level recall, we show in multiple recall tasks (e.g. NIAH-N that requires retrieval of 7 digit numbers from a long context) that this is not the case. CAT’s chunk representations learn to hold information during end-to-end training that helps the decoder to perform fine-grained recall. This is further stress-tested and illustrated by the MQAR experiments (on sequences upto 4x the standard setting, and on the hardest test subset containing maximum key-value tokens to recall) where CAT maintains the fine-grained information despite compression, and outperforms baselines such as GDN and Mamba2 (Figure 5 in the submission).
>
> >  2. CAT appears conceptually similar to NSA [2] without its sparse-attention branch. However, that branch which handles fine-grained long-range interactions, has proven crucial for long-context or information-dense tasks.
>
>
> NSA did not make the observation that _accurate decoding from compressed representations require larger dimensionality_ which we found to be _crucial_ (Table 15 in the appendix shows this ablation).
> Further, note that NSA uses very large chunk sizes of 32.
>
>
> We would like to point out that inference costs in CAT is lower than NSA since NSA requires the full KV cache for the entire sequence for the sparse-attention branch.
>
> > 3. The paper does not specify the exact configuration of the "sparse attention" baseline. If it only uses a static, input-agnostic sparse pattern, the comparison may be unfair, since CAT’s compression involves contextual interaction (via a bidirectional Transformer). For language modeling tasks (that is considered to be semantically rich), dynamic sparse attention methods (e.g., [3]) are generally more representative baselines than static patterns.
>
> We use a fixed strided pattern as described in [5]. The stride is set as 8. We include one more configuration where we set stride as 4.
> We additionally scale up the parameters of the sparse transformers so that it consumes the same memory as CAT for a memory matched apples-to-apples comparison.
>
> Methods such as [3] use a post-hoc KV cache eviction policy, deciding which tokens’ KV cache to keep under a memory budget. However, since models are trained dense but run sparse, train-test mismatch can hurt downstream performance [4]. That being said, we believe [3] to be complementary to any architecture utilizing attention (e.g. hybrid architectures) including CAT since [3] can be applied post-hoc to the dense attention layers in these models after they have been trained.
>
> Note that CAT is trained end-to-end to compress past chunks, resulting in no such mismatch between train and test time. Finally, we would like to note that _not all_ fixed sparse attention masks result in memory savings.
>
> ---
> [1] Dynamic chunking for end-to-end hierarchical sequence modeling, 2025.
>
> [2] Native Sparse Attention: Hardware-Aligned and Natively Trainable Sparse Attention, 2025.
>
> [3] H2O: Heavy-Hitter Oracle for Efficient Generative Inference of Large Language Models, 2023.
>
> [4] Nawrot, Piotr, et al. "The sparse frontier: Sparse attention trade-offs in transformer llms." arXiv preprint arXiv:2504.17768 (2025).
>
> [5] Child et al. 2019, Generating Long Sequences with Sparse Transformers

---

> > ### Author Response · Authors · 2025-11-26
> >
> > ### Questions continued
> >
> > > The performance of CAT 32 is extremely poor on recall tasks. This is likely due to the bottleneck of compressing 32 tokens into one vector. Have you explored alternative compression strategies, such as compressing a chunk of  tokens into  vectors (where ), to provide a better trade-off for larger chunk sizes?
> >
> > Before we arrived at the current architecture, we explored various approaches including compressing into a set of vectors and then decoding from those vectors. However, nothing worked except increasing the dimensionality of the decoder, and hence the parameters of the decoder. With increased parameters of the decoder, even decoding from a single representation worked better than decoding from two vectors but using a less parameter decoder.
> >
> > One could try having multiple vectors with an increased parameter decoder and that may give better quality at the cost of increased inference costs (KV cache would increase 2x due to 2x compressed vectors now). However, we note that decoding from a single compressed representation simplifies the design.

---

### Official Review · Reviewer_KFwk · 2025-10-31

**Soundness:** 4
**Presentation:** 4
**Contribution:** 3
**Rating:** 6
**Confidence:** 3

**Summary:**

The paper proposes Compress & Attend Transformers (CATs): a decoder that attends both to the current local chunk and to compressed representations of prior chunks. CAT trains with multiple chunk sizes so a *single model* exposes a test-time knob trading quality for compute/memory. Experiments show 1.4–3.2× faster generation and 2.2–9.5× lower memory than a dense transformer at similar quality; on in-context recall and long-context tasks, CAT often outperforms efficient baselines (Mamba2, GDN/GDN-Hybrid) and is competitive with dense attention.

**Strengths:**

Originality

* Simple, general recipe: *compress past, attend to compressed past + current chunk*; trains on multiple chunk sizes to enable test-time control without retraining.
* Clear parallel training/generation story; no handcrafted recurrent updates.

Quality

* Provides implementation details (attention mask, KV reuse) and complexity accounting; decoder attention scales as ($O(N^2/C)$).
* Reports broad benchmarks (LM, LongBench, EVAPORATE recall, RULER NIAH), with CAT $>=$ baselines under time/memory-matched settings.

Clarity

* Architecture and training are illustrated nicely.

Significance

* Practical for serving: reduces KV-cache footprint while preserving recall; one model covers multiple quality/compute points.

**Weaknesses:**

1. Mismatched capacity in core tables: Baselines are \~300M params; CAT uses a wider 12-layer decoder + compressor (\~1B). This clouds “architecture vs scale” effects. Please add size-matched CAT (~300M) and/or scaled baselines (\~1B) in the *main* tables.

2. Training cost accounting is thin: The paper notes \~2× longer training but gives no FLOPs/wall-clock/memory breakdown. Add a table with total training FLOPs, hours on fixed hardware, and peak memory for CAT vs baselines.

3. Empirical differentiation from closest work: Related work argues NSA has no inference memory savings (full KV retained), which is CAT’s key advantage, but there’s no direct *peak-memory* comparison on main tasks. Add a head-to-head memory profile vs NSA, and extend the MegaByte/Block comparison beyond a toy task.

**Questions:**

1. Can you report CAT-300M on GovReport/SummScreen and/or ~1B Mamba2/GDN to isolate architecture from scale?
2. On a long-context task (e.g., RULER S-NIAH-U at 4k/8k), report peak inference memory (GB) for CAT vs NSA under matched accuracy.
3. Include results on your LongBench/MQAR setups to substantiate CAT’s claimed recall advantage.
4. What happens if learned compression is replaced with mean/max pooling? Please report impact on perplexity and peak memory.
5. Figure 3 shows 1.4–3.2× speed and 2.2–9.5× memory reductions; could you include per-layer gate/overhead accounting and end-to-end latency vs chunk size?

---

> ### Author Response · Authors · 2025-11-25
>
> ### Weakness 1
>
> > Mismatched capacity in core tables: Baselines are 300M params; CAT uses a wider 12-layer decoder + compressor (1B). This clouds “architecture vs scale” effects. Please add size-matched CAT (~300M) and/or scaled baselines (1B) in the main tables.
>
> We would like to direct the reviewer to our general response ([here](https://openreview.net/forum?id=6rYa2BUnTt&noteId=3r23OwI7TD)) that conveys what the _correct_ comparison criterion should be between architectures. In short, the general response discusses how **raw parameter counts only matter so much as their effect on inference cost, and the right comparison between architectures is what quality one obtains for a given inference cost.** Hence, we compare different models by matching inference costs, rather than raw parameters counts themselves.
>
> _Across varying inference budgets, CAT matches or outperforms different models, all using a single adaptive model only._
>
> Further, for a more exhaustive evaluation and prevent any cherry picking, we pre-trained additional models for each model type by changing their model configurations (scaling up their reccurent state size, or model dimension, or changing their hybrid ratios).
>
> In fact, under same matched inference costs, **CAT outperforms models like `GDN-H-1:7-global-2D` and `Sparse-4/8` that use similar parameters as CAT.** More importantly, note that these are different models trained from scratch that occupy different locations on the quality-inference cost curve and CAT outperforms them using just a single model.
>
> Finally, we would like to highlight (see [general response](https://openreview.net/forum?id=6rYa2BUnTt&noteId=3r23OwI7TD)) that CAT can be thought of as a **meta sequence mixer** i.e. general arrangement of compressor and decoder blocks. The compressor and decoder themselves could be made up of any architecture. This makes CAT complementary to existing efficient architectures and all the excellent research being done on efficient architectures can be readily used in CAT. Additionally, CAT can be instantiated as a layer, which can be used as a drop-in replacement layer in any architecture. This unlocks interesting possibilities of new generation of _hybrid as well as natively adaptive_ architectures

---

> > ### Author Response · Authors · 2025-11-25
> >
> > ### Weakness 2
> >
> > > 2. Training cost accounting is thin: The paper notes ~2× longer training but gives no FLOPs/wall-clock/memory breakdown. Add a table with total training FLOPs, hours on fixed hardware, and peak memory for CAT vs baselines.
> >
> > We report both empirical system-level measurements and theoritical per-layer FLOP complexity for CAT and the dense transformer baseline.
> >
> > Below table provides FLOPs used per layer.
> >
> > | Component       | CAT                              | dense transformer  |
> > | --------------- | -------------------------------- | -------------------------- |
> > | Training Attention FLOPs | $O(\frac{N^{2}}{C})$ | $O(N^{2})$     |
> > | Training MLP FLOPs       | $O(4ND^2)$       | $O(ND^2)$   |
> >
> >
> > Below table provides training throughput comparison.
> >
> > | Model | Training peak memory (GB) | Training throughput (tokens/sec) |
> > | ----- | ---------------- | ----------------------- |
> > | CAT   | 37.3           | 85K                |
> > | Dense | 14.0          | 203K                |
> >
> > We used an A100 80GB PCIe GPU for these training throughput and memory measurements.
> >
> > Further, note that CAT uses pure PyTorch implementation meaning these throughput numbers for CAT are a lower bound on how fast CAT can be trained.
> >
> > Moreover, this 2x increase in training time is largely an artifact of _inefficiencies in FlexAttention API by PyTorch rather than the CAT architecture itself_. FlexAttention API does not currently exploit the specific sparsity in attention mask used by CAT during training [1]. Theoretically, CAT reduces the attention FLOPs by a factor of C (chunk size) compared to a causal dense attention, meaning there is a _potential for a significant speed up in pre-training_. We expect a dedicated kernel that exploits this sparsity to substantially reduce this gap, and is left as a future work.
> >
> > Importantly, we note that CAT amortizes this overhead by effectively training multiple models in a single pre-training run (corresponding to different inference budgets). To obtain an equivalent set of models with dense transformers or any other efficient model, one would need to train multiple separate models from scratch, which would require substantially more total compute than our single CAT training run – even accounting for the current suboptimal ~2× training-time increase. From this perspective, CAT reduces overall training cost for a multi-inference budget deployment.
> >
> > [1] Wang et al. 2024, FlashMask: Efficient and Rich Mask Extension of FlashAttention

---

> > > ### Author Response · Authors · 2025-11-25
> > >
> > > ### Weakness 3
> > >
> > > > Empirical differentiation from closest work: Related work argues NSA has no inference memory savings (full KV retained), which is CAT’s key advantage, but there’s no direct peak-memory comparison on main tasks. Add a head-to-head memory profile vs NSA
> > >
> > > We would like to note that NSA will consume slightly more memory as a comparable dense transformer due to: (i) KV cache (which is same in both NSA and dense) and the (ii) extra compressed KV cache in NSA. This means one can consider a dense transformer as a lower bound memory profile for NSA. Memory used per task depends on the maximum context length used for that task (real-world in-context recall tasks use upto 2K whereas NIAH-N uses upto 4K sequences). We have provided how the memory grows in different architectures in Figure 3 as the sequence length increases for different architectures. Here, one can consider dense transformer curve as a proxy for NSA.
> > >
> > > > extend the MegaByte/Block comparison beyond a toy task
> > > We used the synthetic MQAR task to probe different architectures on their in-context recall performance.
> > >
> > > It has been observed consistently in the literature that MQAR task has a very-high direct correlation with downstream recall performance on real-world tasks [1, 2].
> > > Due to limited resources, we were not able to run Block Transformer beyond MQAR. However, if the reviewer feels this is very important, we can schedule a pre-training run for this baseline too. Let us know :)
> > >
> > > [1] Arora et al. Simple linear attention language models balance the recall-throughput tradeoff
> > >
> > > [2] Allen-Zhu et al. Physics of Language Models: Part 4.1, Architecture Design and the Magic of Canon Layers

---

> ### Author Response · Authors · 2025-11-25
>
> ### Questions
>
> > 1. Can you report CAT-300M on GovReport/SummScreen and/or ~1B Mamba2/GDN to isolate architecture from scale?
>
> We would like to direct the reviewer on our general response 1 [here](https://openreview.net/forum?id=6rYa2BUnTt&noteId=3r23OwI7TD) where **we compare architectures on the _correct_ matching criterion, which is inference costs and not parameters**. There we observe CAT is at the pareto frontier of quality-inference costs trade-off curve.
>
> Further, we report scaled up versions of different model types in our general response [here](). In fact, under same matched inference costs, **CAT outperforms models like `GDN-H-1:7-global-2D` and `Sparse-4/8` that use similar parameters as CAT.** More importantly, these are different models trained from scratch that occupy different locations on the quality-inference cost curve and CAT outperforms them using just a single model.
>
>
> > 2. On a long-context task (e.g., RULER S-NIAH-U at 4k/8k), report peak inference memory (GB) for CAT vs NSA under matched accuracy.
>
> Since NSA closely tries to match dense transformer performance [1], we consider dense transformers as a proxy for NSA.
> While we don’t have a result where the accuracy matches exactly, we have a result where CAT-8 interestingly outperforms the dense transformer on S-NIAH-U at 4K context length (47.3 vs 19.8). This translates to using 4x lower memory than a dense transformer (or NSA) while at the same time, having a higher accuracy than dense transformer (or NSA).
>
> [1] Yuan et al. (2025), Native Sparse Attention: Hardware-Aligned and Natively Trainable Sparse Attention
>
> > 3. Include results on your LongBench/MQAR setups to substantiate CAT’s claimed recall advantage.
>
> We would like to direct the reviewer to Table 3 and Figure 4 where we report results on LongBench and the MQAR task respectively in our submission.
>
> > 4. What happens if learned compression is replaced with mean/max pooling? Please report impact on perplexity and peak memory.
>
> We would like to thank the reviewer for their suggestions. We plan to add this ablation in our final manuscritpt where learned compression is replaced with a mean pooling function (instead of a transformer performing compression).
>
> We note that the memory during inference won’t change even if learned compression is replaced with mean pooling operation since memory during inference is controlled by the KV cache of the decoder (which is independent of how the compression happens).
>
> > 5. Figure 3 shows 1.4–3.2× speed and 2.2–9.5× memory reductions; could you include per-layer gate/overhead accounting and end-to-end latency vs chunk size?
>
> We report breakdown of time taken by each component of CAT during generation. We use chunk size of 8 for this ablation.
>
> | Component                 | Time (ms)  | Percentage |
> | ------------------------- | ---------- | ---------- |
> | Attention in decoder      | 30,817     | 70.1       |
> | FFN in decoder            | 11,555     | 26.3       |
> | Compression in compressor | 1,551      | 3.5        |
> | **Total**                 | **43,932** | **100.0**  |
>
> We observe that attention in CAT decoder dominates the runtime, accounting for roughly 70% of total time, followed by the FFN layers at about 26%. In contrast, the parallel compression in the CAT compressor contributes only ~3.5% of the overall runtime, indicating that its overhead is negligible compared to the main decoding computation.

---

### Official Review · Reviewer_7wBq · 2025-11-03

**Soundness:** 2
**Presentation:** 3
**Contribution:** 3
**Rating:** 4
**Confidence:** 4

**Summary:**

Many methods a-priori constrain the memory and compute use of a model, or methods that recursively compress context over time tend to be slow and difficult to optimize. At the same time, different tasks and contexts face different compute and memory requirements. In response, this work introduces CAT to compress chunks of tokens at a time.

**Strengths:**

- The problem is well-defined and important
- The idea of training with multiple chunk sizes to enable an adaptive model is useful and interesting, given that most efficient architectures choose the memory and compute budget a-priori
- The results in Table 3 and Figure 3 are very interesting; for a similar latency to Mamba-2/GatedDeltaNet, CAT-8 offers much better quality. It’s also interesting that a single model is used for all the experiments, highlighting that the compute-quality can be tuned.

**Weaknesses:**

The parameter scales differ substantially (CAT ≈ 1 B vs. 300 M baseline), making it difficult to isolate the contribution of the architecture. Apples-to-apples scaling ablations would strengthen the case. Especially at such small parameter scales, it’s difficult to understand these trends.

I will certainly consider increasing my score if I can better understand the impact of the parameter scale differences.

**Questions:**

- How does the proposed chunk-based compression approach compare to (1) log-linear attention, (2) methods that encode chunks of text using a retriever/encoder model and then use similarity-based metrics to select the most relevant chunks for decoding?

- What happens to tokens on a chunk boundary? Some prior work uses sliding window attention to capture full attention for local tokens in the sequence. If a token is right past the prior chunk boundary, does it get to fully attend to the lagging few tokens, or does it only attend to the compressed version of the tokens? It would be interesting to see ablations for the boundary token perplexity vs. perplexity of tokens later within chunks that get to attend to a lot of recent tokens.

- It’s surprising that GDN-Hybrid and the dense transformer model perform so poorly on FDA and NIAH 4K. It would be helpful to understand from the error analysis what the authors observe.

- It is surprising that the method works well on MQAR/NIAH since those tasks have very fine-grained keys and values without many distractor tokens (especially for MQAR with many keys and values); it seems like even compressing 4 tokens could drop a lot of precision. How do you think the chunk representation encompases all the keys and values in its chunk with high fidelity? It would be interesting to perform MQAR experiments as you scale up the number of keys and values in the sequence.

---

> ### Author Response · Authors · 2025-11-25
>
> ### Weakness 1
> > The parameter scales differ substantially (CAT ≈ 1 B vs. 300 M baseline), making it difficult to isolate the contribution of the architecture. Apples-to-apples scaling ablations would strengthen the case. Especially at such small parameter scales, it’s difficult to understand these trends.
> I will certainly consider increasing my score if I can better understand the impact of the parameter scale differences.
>
> We would like to direct the reviewer to our general response 1 [here](https://openreview.net/forum?id=6rYa2BUnTt&noteId=3r23OwI7TD), where we make an _apples-to-apples comparison between different architectures_. Specifically, we note that **raw parameter counts only matter so much as their effect on inference cost, and the right comparison is _what_ quality a model obtains for a given inference cost.** Hence, we compare different models by _matching inference costs_. Further, for a more exhaustive evaluation and prevent any cherry picking, we pre-trained additional models for each model type by changing their model configurations (scaling up their reccurent state size, or model dimension, or changing their hybrid ratios). In total, this results in around 12 models that we compare CAT against. These models have a range of parameter counts, from ~300M to upto ~800M, with varying inference costs. Every model type has atleast ≥2 model configurations to ensure fair and broader evaluation. These models are different points on the quality-inference costs trade-off curve.
>
> We observe that **CAT obtains a pareto frontier in the quality-inference costs trade-off curve across all models, including the scaled up ones. Importantly this is achieved using just a single model.**
>
> In fact, under matched inference costs, **CAT outperforms models like `GDN-H-1:7-global-2D` and `Sparse-4/8` that use similar parameters as CAT.** Note that these, including others are different models that occupy different locations on the quality-inference cost curve. Any prior baseline requires retraining from scratch to target different inference budgets, whereas **CAT makes it easy to _traverse_ this trade-off directly at test-time.**

---

> > ### Author Response · Authors · 2025-11-25
> >
> > ### Questions
> >
> > > How does the proposed chunk-based compression approach compare to (1) log-linear attention, (2) methods that encode chunks of text using a retriever/encoder model and then use similarity-based metrics to select the most relevant chunks for decoding?
> >
> > (1) Recently introduced log linear attention (parallel submission in this conference) alleviates the fixed memory bottleneck from linear attention. To compare CAT with log linear attention (using Mamba2 as the base linear attention mixer), we use the MQAR task. We used the implementation provided here: [flash-linear-attention repo](https://github.com/fla-org/flash-linear-attention/tree/main/fla/ops/log_linear_attn).
> > However, after trying multiple different configurations (sweep over layers, state size, hidden size) and sweeping over learning rates, the best accuracy we could obtain was 0.03%. The MQAR task setting used is same as desribed in Figure 5 in our submission.
> >
> > However, we observe that log linear variants perform only marginally better compared to their standard linear only versions (Figure 5 in [1]). If this trend extrapolates to longer sequences of upto 1024, we do expect CAT to outperform log linear attention while taking even lower memory (see Figure 5 in our submission; additionally [1] tried the MQAR task on sequence lengths upto 256 only, unlike our setting that goes upto 1K sequence lengths).
> >
> > [1] Guo et al. 2025, Log-Linear Attention
> >
> > (2) By encoder-based retrieval and then similarity selection of chunks for decoding, we believe the reviewer is thinking about RAG-style or retrieval-augmented language models [2, 3]. These works usually use a pretrained encoder to encode a very large text corpus (this is pre-computed and maintained as a retrieval index), and then during inference, “similar looking” chunks to the input text are retrieved and inserted to the input context before performing generation. We believe these works are complementary since one can perform RAG on top of CAT. In fact, due to compression of the context, CAT would speed up generation, and further enable more chunks to be retrieved from the corpus since CAT compresses the context.
> >
> > [2] Lewis, Patrick, et al. "Retrieval-augmented generation for knowledge-intensive nlp tasks." Advances in neural information processing systems 33 (2020): 9459-9474
> >
> > [3] Borgeaud, Sebastian, et al. "Improving language models by retrieving from trillions of tokens." International conference on machine learning. PMLR, 2022.
> >
> > > What happens to tokens on a chunk boundary? Some prior work uses sliding window attention to capture full attention for local tokens in the sequence. If a token is right past the prior chunk boundary, does it get to fully attend to the lagging few tokens, or does it only attend to the compressed version of the tokens?
> >
> > If a token is right past the prior chunk boundary, it **only attends to the compressed version of the tokens** as reflected in the schematic diagram of CAT architecture (Figure 2 of our submission).
> >
> >
> > > It would be interesting to see ablations for the boundary token perplexity vs. perplexity of tokens later within chunks that get to attend to a lot of recent tokens.
> >
> > We provide how the validation loss changes within a chunk. [Here](https://ibb.co/zTN6Mx37) is a link for the figure. We provide averaged results across all chunks.
> > We provide different curves for each chunk size.
> > Interestingly, across all chunk sizes, the loss is highest when decoding the first token from the compressed representation only.
> > After that token is decoded, the loss decreases steadily as CAT keeps decoding tokens from both compressed representation and raw tokens that appear before inside the chunk.
> >
> > We will add this analysis in our final manuscript.

---

> > > ### Author Response · Authors · 2025-11-25
> > >
> > > ### Questions continued
> > >
> > > > It’s surprising that GDN-Hybrid and the dense transformer model perform so poorly on FDA and NIAH 4K. It would be helpful to understand from the error analysis what the authors observe.
> > >
> > > We provide some responses of GDN-Hybrid and dense transformers on NIAH and FDA.
> > >
> > >
> > > Consider a random sample from NIAH-N 4K task where the model needs to retrieve the needle number `2338687` from the long-context. Here are the responses for each model (we generate upto 42 tokens for each sample):
> > >
> > > ```bash
> > > Dense (does not generate any number):
> > > the magic number for the number of people who can be reached by a single click. It's the magic number for the number of people who can be reached by
> > >
> > > GDN-Hybrid (hallucinates the number):
> > > 5.0. The special magic number for harmonious-reduction mentioned in the provided text is 5.0. The special magic number for harmonious-
> > >
> > > CAT-8 (generates the correct number):
> > > 2338687. I'm not sure if this number is a magic number, but it's a good one. I think it's a good one.
> > >
> > > ```
> > >
> > > Here is another sample for the same task where model needs to retrieve the needle number `5479144`
> > >
> > > ```bash
> > > Dense (does not generate any number):
> > > the number that is the most likely to be true. The number is the number that is the most likely to be true. The number is the number that is
> > >
> > > GDN-Hybrid (gets the first digit correct, but not the rest of the number):
> > > 5x a year. The special magic number for uncovered-hypothesis mentioned in the provided text is 5x a year. The special magic number for uncovered
> > >
> > > CAT-8 (generates the correct number):
> > > 5479144. The magic number is 5479144. The magic number is 5479144. The magic number is 5479144. The magic number is
> > >
> > > ```
> > >
> > > Here is a sample on the FDA task. The model needs to retrieve the answer: `Microbiology (83)` from the context.
> > >
> > > ```bash
> > > Dense:
> > > Molecular Biology
> > > The ARIES Bordetella Assay is a real-time PCR assay that detects Bordetella pertussis and …
> > >
> > > GDN-Hybrid:
> > > The ARIES Bordetella Assay is a real-time polymerase chain reactio …
> > >
> > > CAT-8:
> > > Microbiology (83) H. Intended Use: 1. Purpose: Intended Use: The ARIES Bordetella Assay is a real-time polymerase …
> > > ```
> > >
> > > Interestingly, the line `The ARIES Bordetella Assay is a real-time ...` is very close to where the actual answer text `Microbiology (83)` is. Perhaps, both Dense and GDN-H1 got distracted and instead retrieved the wrong answer text.
> > >
> > > Here is one more sample from FDA task where the model needs to retrieve `Class II`.
> > >
> > > ```bash
> > > Dense:
> > > 1. Class I: 1.1 ….
> > >
> > > GDN-Hybrid:
> > > 4-2-2 and 5-3-3 Configurations BD FACS 7-Color Setup BD Multi-Check Control
> > >
> > > CAT-8:
> > > Class II 3. Product code: OYE, flow cytometric …
> > > ```

---

> ### Author Response · Authors · 2025-11-25
>
> ### Questions continued
>
> > It is surprising that the method works well on MQAR/NIAH since those tasks have very fine-grained keys and values without many distractor tokens (especially for MQAR with many keys and values); it seems like even compressing 4 tokens could drop a lot of precision. How do you think the chunk representation encompases all the keys and values in its chunk with high fidelity? It would be interesting to perform MQAR experiments as you scale up the number of keys and values in the sequence.
>
> The reviewer’s intuition is correct that compression throws away information. However, whether one can determine the correct value from a compressed context depends both on the degree of compression and the vocabulary size. As an extreme example, consider a vocabulary of size 256 where the elements are encoded as binary vectors of size 32. One only needs 8 bits to represent any element in the vocabulary, allowing for lossless compression of 4 different elements into one 32-bit representation, allowing for perfect recall.
>
> In the experiment in the paper, CAT uses a continuous embedding (which is more expressive than a binary one) of dimension of 128, which allows for representing exponentially more vectors than the possible 8192 vocabulary elements in the MQAR vocabulary.
>
> Further, whether an architecture _learns_ to hold this information for accurate prediction, even if sufficient representation size is provided is an empirical question, and CAT does learn to hold information for accurate decoding, perhaps due to end-to-end learning.  **Figure 5 in the submission stress-tests CAT and other baselines on the MQAR task, with the maximum number of key-values that can fit on sequence lengths upto 1K (which is 4x the standard sequence length used for this task)**. CAT with a chunk size of 4 is able to recall every key-value pair almost perfectly, outperforming baselines such as GDN and Mamba2 especially at longer contexts.

---

### Official Review · Reviewer_Ue8h · 2025-11-04

**Soundness:** 1
**Presentation:** 3
**Contribution:** 2
**Rating:** 0
**Confidence:** 5

**Summary:**

The paper introduces the Compress and Attend Transformer (CAT), an architecture intended to be a controllably efficient alternative to standard transformers. It processes input by dividing it into chunks. When decoding a new token, the model attends to a local window of nearby tokens and also to a sequence of single vector compressed representations of all past chunks. The level of efficiency is controlled at test time by choosing the chunk size (C); larger chunks mean more compression and lower computational cost. The paper also presents a method to train a single, adaptive model that can operate with multiple chunk sizes. The authors claim this single model outperforms other efficient baselines and is 1.5-3x faster and uses 2-9x less memory than a dense transformer.

**Strengths:**

1. The core concept of a single, adaptive model that can be adjusted at test time to trade quality for compute is a novel and potentially useful contribution. This provides a flexibility that is absent in most existing efficient architectures, which are fixed into a specific configuration at training time.

2. The paper is clearly written. The central mechanism of the CAT architecture is explained well and is easy to grasp, particularly with the help of Figure 2, which provides a simple visual of the information flow.

3. The goal of the paper is significant. A single model that could be deployed to serve a wide range of needs (from low latency chat to high recall analysis) would be a major practical advantage. However, whether the paper actually achieves this significance is highly questionable due to major methodological flaws.

**Weaknesses:**

1. The primary weakness is the invalid comparison in the main results tables (Tables 2, 3, 4, 5). The proposed CAT model has approximately 1B parameters. This model is compared against baselines (Dense, Mamba2, GDN) that have only 300M parameters. This 3x-4x parameter disparity invalidates any claims of superior efficiency or performance. The CAT model is not a more efficient architecture; it is a much larger model.

2. The paper's own parameter matched comparison, in Appendix A.3 (Table 7), directly contradicts the main claims from the abstract. When CAT is compared to a "Dense 2D" model of similar size, the Dense 2D model is superior on all language modeling and common sense reasoning tasks. The CAT model only wins on two specific recall datasets, suggesting the architecture is not a general improvement but rather a specialized design that trades general performance for recall. This is a critical finding that is omitted from the main paper, making the abstract's claims misleading.

3. The headline claims of being "1.5-3x faster and requiring 2-9x lesser memory"  are made against a 300M dense transformer, not a parameter matched one. A 1B parameter model should be compared to a 1B parameter baseline. The paper notes CAT-4 is 3x faster than a parameter matched dense model (Dense 2D) , but this is a much less impressive claim when Table 7 shows it's also a worse model on standard benchmarks.

4. The authors admit the CATS model takes "twice as much time" to train as the (much smaller) dense baseline and up to 2.35x longer than a dense model of equivalent decoder depth. This is a significant practical disadvantage that is dismissed as a "one time cost" but represents a major barrier to adoption.

**Questions:**

1. The main tables (e.g., Table 2, 3) compare a 1B parameter CAT to 300M parameter baselines. Your own appendix (Table 7) shows that a parameter matched "Dense 2D" model outperforms CAT on all language modeling and common sense reasoning tasks. How can the abstract's claims of outperforming baselines be justified when your own data contradicts this?

2. The paper states that a 2x wider decoder ($D_g = 2D$) is required to match perplexity. This suggests the CAT architecture itself is less parameter efficient than a standard transformer. What is the performance of a CAT model that has the same total parameter count as the 300M dense baseline (e.g., a 12 layer model with $D=1024$ split between the compressor and decoder)?

3. The performance of CAT 32 is extremely poor on recall tasks. This is likely due to the bottleneck of compressing 32 tokens into one vector. Have you explored alternative compression strategies, such as compressing a chunk of $C$ tokens into $k$ vectors (where $1 < k < C$), to provide a better trade-off for larger chunk sizes?

---

> ### Author Response · Authors · 2025-11-26
>
> ### Weaknesses
>
> > The primary weakness is the invalid comparison in the main results tables (Tables 2, 3, 4, 5). The proposed CAT model has approximately 1B parameters. This model is compared against baselines (Dense, Mamba2, GDN) that have only 300M parameters. This 3x-4x parameter disparity invalidates any claims of superior efficiency or performance.
>
> The reviewer seems to suggest that it is only ever valid to compare parameter-matched models despite the differences in the inference costs. We disagree.
>
> We would like to direct the reviewer to our general response 1 [here](https://openreview.net/forum?id=6rYa2BUnTt&noteId=3r23OwI7TD), where we make an _apples-to-apples comparison between different architectures_. To summarize, **raw parameter counts only matter so much as their effect on inference cost, and it is better to compare the quality a model obtains for a given inference cost.** Hence, we compare different models by _matching inference costs_.
>
> Further, for a more exhaustive evaluation and to prevent any cherry picking, we pre-trained additional models for each model type by changing their model configurations (scaling up their recurrent state size, or model dimension, or changing their hybrid ratios). In total, this results in around 12 models that we compare CAT against. These models have a range of parameter counts, from ~300M to upto ~800M, with varying inference costs. Every model type has atleast ≥2 model configurations to ensure fair and broader evaluation. These models are different points on the quality-inference costs trade-off curve.
>
> > **In the attached figure [here](https://ibb.co/d49gBVTH),** we plot the quality vs inference costs for different models as well as CAT.
>
> **CAT obtains a pareto frontier in the quality-inference costs trade-off curve across all models, including the scaled up ones. Importantly this is achieved using just a single model.**
>
> In fact, under matched inference costs, **CAT outperforms models like `GDN-H-1:7-global-2D` and `Sparse-4/8` that use similar parameters as CAT.** Note that these, including others are different models that occupy different locations on the quality-inference cost curve. Any prior baseline requires retraining from scratch to target different inference budgets, whereas **CAT makes it easy to _traverse_ this trade-off directly at test-time.**

---

> ### Author Response · Authors · 2025-11-26
>
> > The CAT model is not a more efficient architecture; it is a much larger model.
>
> and
>
> > The paper's own parameter matched comparison, in Appendix A.3 (Table 7), directly contradicts the main claims from the abstract. When CAT is compared to a "Dense 2D" model of similar size, the Dense 2D model is superior on all language modeling and common sense reasoning tasks. The CAT model only wins on two specific recall datasets, suggesting the architecture is not a general improvement but rather a specialized design that trades general performance for recall. This is a critical finding that is omitted from the main paper, making the abstract's claims misleading.
>
> and
>
> > The headline claims of being "1.5-3x faster and requiring 2-9x lesser memory" are made against a 300M dense transformer, not a parameter matched one. A 1B parameter model should be compared to a 1B parameter baseline. The paper notes CAT-4 is 3x faster than a parameter matched dense model (Dense 2D) , but this is a much less impressive claim when Table 7 shows it's also a worse model on standard benchmarks.
>
> We would like to direct the reviewer to our second general response 2 ([here](https://openreview.net/forum?id=6rYa2BUnTt&noteId=WgvmGO1piP)), where we discuss exactly this concern.
>
> In short, Dense-2D is significantly more expensive to run when compared with other efficient architectures (e.g., GDN-Hybrid) including CAT.
> From this inference-cost perspective, Dense 2D is in fact an unfair comparison to CAT or any other efficient baseline for that matter.
> Our claims are grounded in comparisons at fixed inference costs, where **CAT matches dense transformer performance while still being efficient in inference costs**, **and parameter count matter only so much as they affect the inference cost.** We reported Dense 2D performance for _transparency_ and _completeness_ of the results, and NOT as a direct comparison with CAT.
>
> To enable a fair comparison, one would scale down a dense transformer such that it is memory matched to CAT-4 (either by reducing hidden size, or decreasing number of layers). This dense transformer then would consume the same memory as CAT-4, but can only be worse than CAT-4 on real world recall tasks and language modeling.
>
> Further, when actually comparing Dense D with CAT-4, we note CAT-4 consumes 2x less memory and generates 1.5x faster. At the same time, CAT-4 matches Dense D on language modeling evaluations. Hence, the claim that CAT matches a dense transformer on language modeling while being more efficient is valid.
>
> > A 1B parameter model should be compared to a 1B parameter baseline.
>
> The reviewer’s suggestion does not necessarily help address concerns about efficiency.
>
> To motivate a different comparison (based on the quality at an inference budget), consider the example of Mixture-of-Experts (MoEs). Models with MoE layers are now widely deployed everywhere despite having 10–20× more parameters. When compared at equal total parameter counts, MoEs underperform compared to dense transformers (Table 2 in [1]). The key improvement is the routing-based computational structure.
>
> CAT should be viewed in this same spirit: by using compressed computation, CAT achieves competitive or better performance than dense transformers while being more efficient.
>
> > The headline claims of being "1.5-3x faster and requiring 2-9x lesser memory" are made against a 300M dense transformer
>
> To motivate our comparison, we would like to ask the reviewer to consider this question from our general response:
> **Suppose model A (CAT having 1B parameters) has more parameters than model B (Dense having 300M parameters), then if model A outperforms model B using lower inference costs (1.5-3x faster and requiring 2-9x lesser memory), does it matter that model A has more parameters than model B? Which model should one deploy?**
>
> [1] Dai et al. 2024, DeepSeekMoE: Towards Ultimate Expert Specialization in Mixture-of-Experts Language Models

---

> > ### Author Response · Authors · 2025-11-26
> >
> > > The authors admit the CATS model takes "twice as much time" to train as the (much smaller) dense baseline and up to 2.35x longer than a dense model of equivalent decoder depth. This is a significant practical disadvantage that is dismissed as a "one time cost" but represents a major barrier to adoption.
> >
> > We respectfully disagree with the characterization that the increased training time constitutes a “major barrier to adoption.”
> >
> > Firstly, the reported ~2× increase in training time is a linear, not exponential, overhead and is well within the range of training-budget variation that is common when adopting new architectures. Moreover, this factor is largely an artifact of inefficiencies in FlexAttention API by PyTorch rather than CAT architecture itself. FlexAttention API does not currently exploit the specific sparsity in attention mask used by CAT during training [1]. Theoretically, CAT reduces the attention FLOPs by a factor of C (chunk size) compared to a causal dense attention, meaning there is a potential for a significant speed up in pre-training. We expect a dedicated kernel that exploits this sparsity to substantially reduce this gap, and is left as a future work.
> >
> > Secondly, CAT amortizes this overhead in training by effectively training multiple models in a single pre-training run (corresponding to different inference budgets). To obtain an equivalent set of models with dense transformers or any other efficient model, one would need to train multiple separate models from scratch, which would require substantially more total compute than our single CAT training run – even accounting for the current suboptimal ~2× training-time increase. From this perspective, CAT reduces overall training cost for a multi-inference budget deployment, rather than increasing it.
> >
> > [1] Wang et al. 2024, FlashMask: Efficient and Rich Mask Extension of FlashAttention

---

> ### Author Response · Authors · 2025-11-26
>
> ### Questions
>
> > The paper states that a 2x wider decoder is required to match perplexity. This suggests the CAT architecture itself is less parameter efficient than a standard transformer.
>
> We would again like to emphasize that parameter counts themselves do not dictate inference costs of architectures. While we agree having a parameter efficient architecture is a useful scientific endeavor, we note that deployment of architectures in the real world does not pay any heed to parameter counts.
>
> A great example of this is Mixture-Of-Experts, they are now part of every flagship LLM that is deployed in the real world, and are deliberately over parameterized than dense transformers (upto 10-20x more) while having lower inference costs. If one compares an MoE with a dense transformer of the same total parameters, MoE model underperforms compared to the dense transformer [1]. Is the MoE model by this rationale of “less parameter efficient” not an efficient architecture and not useful as an architecture?
>
> Further, we agree with the reviewer that using a 2x wider decoder in CAT is required to match perplexity. For an ablation on this, refer to Table 15 in the appendix. With this ablation, we conclude that if one wants to decode accurately from a compressed representation, one requires higher dimensionality during decoding.
>
> This interplay between higher dimensionality required to decode accurately from compression can be seen in recent works [2]. ([2] claims that the optimal way to have best quality-inference cost trade-off in vision language models is to have larger models and highly compressed images, even compressed to just one token. Lower inference costs come from reduced sequence length while still using a substantially larger parameter model. This is exactly similar to CAT).
>
> [1] Dai et al. 2024, DeepSeekMoE: Towards Ultimate Expert Specialization in Mixture-of-Experts Language Models
>
> [2] Goyal et al. 2024, Inference Optimal VLMs Need Fewer Visual Tokens and More Parameters

---

### Author Response · Authors · 2025-11-25
**General Response 1: comparing models under matched inference costs, and not parameter counts**

Several reviewers asked why CAT models were not "parameter matched" to the baselines and suggested that the differences in parameter counts make the comparison unfair.

In this general response, we argue why the raw parameter count is **not** the appropriate matching criterion.

### Inference costs matter, not parameters

**Inference costs dominate:** Major costs of language models include (i) training these models, and then (ii) serving these models for inference.  While both training and inference costs of these models are significant and important, the cost of inference _dominates_ in the long run when deploying language models in the real world [1, 2, 3, 4, 5, 6]. In fact, the costs of training can break even with inference in _just_ a few weeks after deployment [6]. Further, this time to break even is decreasing rapidly as more people are using reasoning language models with increasing context lengths.


**Raw parameter counts alone do not determine inference costs:** Now, inference costs depend on the time that a model takes to process/generate a sequence (computations required to perform forward passes) and the hardware/GPUs required to process/generate sequences (the memory required to run the model). **The raw parameter count affects the inference cost only insofar as it affects computations required and memory usage.** As an example, consider Mixture-of-Experts (MoEs), which are now the default choice in all flagship LLMs [8, 9]. MoEs are _explicitly_ designed to have more parameters (>10-20x more [8, 9]) than a dense transformer while keeping inference costs essentially unchanged [10].

**As a result, what matters for model deployment is the quality at a desired inference cost, and not the raw parameter count itself.**

To motivate this idea of quality at an inference costs further, we would like to ask the reviewers:
**Suppose model A has more parameters than model B, then if model A outperforms model B using lower inference costs, does it matter that model A has more parameters than model B? Which model should one deploy: model A or B?**

In a similar vein to MoEs, despite using more parameters, **CAT gives better quality while using similar inference costs as other models.**

> Before moving on, we note that inference cost depends on both the underlying algorithm and its implementation on hardware (Python/PyTorch or CUDA/Triton code). However, CAT is implemented using simple pure PyTorch code, meaning the improvements in costs are not due to a smart implementation, but due to the CAT architecture itself (i.e. compress and attend): meaning CAT can be made even faster by the community. This is unlike the models we compare with, where most have optimized CUDA/Triton code.

---

> ### Author Response · Authors · 2025-11-25
>
> ### Comparing models with what quality is obtained at a given inference cost
>
> Keeping the above discussion in mind, we compare different models by measuring what quality they get for a given inference cost, specifically memory usage, since it’s the major inference time bottleneck in increasingly memory-bound GPU workloads [12]. We measure quality using real-world in-context recall tasks.
>
> In our submission, we picked standard baselines from the literature [11] at their default hyperparameter settings. To aid in further fair evaluation of different models and prevent any _cherry-picking_, **we run seven additional pre-training runs. Some were suggested by the reviewers (McFn) and some are added by us to enable a broader comparison.**
>
> Specifically, **for _each model type_, we pre-train a new model from scratch by scaling it up and/or changing its model configuration.** Following are the changes we made for each model type to enable a fair comparison to CAT:
> - GDN-Hybrid
>     - `GDN-H-1:7-global`: Change hybrid ratios of GDN-Hybrid to 1:7 with global attention (suggested by reviewer McFn)
>     - `GDN-H-1:7-global-2D`: Change hybrid ratios of GDN-Hybrid to 1:7 with global attention, and increasing model dimension by 2x (suggested by reviewer McFn) –- **note that this model has a similar number of parameters as CAT**
>     - `GDN-H-1:4`: Change hybrid ratios of GDN-Hybrid to 1:4
>     - `GDN-H-1:1-D/2`: Change the model dimension of GDN-Hybrid 1:1 (reduce by 2x)
> - GDN
>     - `GDN-2x`: Increase recurrent state size of GDN to 2x
> - Dense
>     - `Dense-D/2`: Change the model dimension (reduce by 2x)
> - Sparse
>     - `Sparse-4`: Decrease the sparsity ratio by 2x, and increase model dimension by 2x –- **has a similar number of parameters as CAT**
>
> In total, this results in around **12 models** that we compare CAT against. These models have a range of parameter counts, from ~300M to upto ~800M, with varying inference costs. Every model type has atleast ≥2 model configurations to ensure fair and broader evaluation. These models appear at different points on the quality-inference costs trade-off curve.
>
> > **In the attached figure [here](https://ibb.co/d49gBVTH),** we plot the quality vs inference costs for different models as well as CAT.
>
> We observe that CAT has good performance as inference costs increase, achieving almost a pareto frontier in quality (in-context recall) vs inference cost (memory usage). **In other words, CAT gives equal or better quality when _matched_ for the same inference costs across different budgets.**
>
> **More importantly, CAT achieves this pareto-frontier using a _single model only_, whereas _prior_ approaches would require re-training from scratch to target different inference budgets.**
>
> We have updated the corresponding Figure in the submission.

---

> > ### Author Response · Authors · 2025-11-25
> >
> > ### CAT provides a _knob_ to control efficiency
> >
> > One can move in this quality-efficiency trade-offs (by changing model configuration e.g. changing hybrid ratios, or increasing recurrent state size or model dimension etc.). Any prior approach will require separate models trained from scratch to target different inference budgets, that lead to different grey points in the quality-inference cost curve (see figure [here](https://ibb.co/d49gBVTH)).
> >
> > However, CAT allows one to _traverse_ along this trade-off at test-time without any training -- **a capability absent in most efficient architectures**, as reviewers #Ue8h and #7wBq point out.
> >
> > **_More importantly, this need to carefully tune model configuration (e.g. tuning hybrid ratios to 1:1, 1:7 or any other 1:X etc.) for a desired quality-efficiency trade-off is precisely what CAT is designed to avoid at scale_: CAT exposes multiple trade-offs directly at test time without retraining.**
> >
> > ### CAT is a _meta_ sequence mixer
> >
> > CAT can be best described as a _"meta sequence mixer"_ and is complementary to existing efficient sequence mixers.
> > CAT has two components: a compressor and a decoder – each of these could make use of any sequence mixers, such as linear attention.
> >
> > We provide a preliminary result on the MQAR task where the decoder in CAT is a GDN-Hybrid architecture (having a 1:1 ratio). We test on sequence lengths of upto 256 where the number of keys-values are maximum.
> >
> > This new architecture solves this task, empirically demonstrating the use of GDN layers inside of CAT: meaning rather than CAT being a strict competitor to GDN (or any other efficient sequence mixer), **CAT is complementary to existing sequence mixers.** _Further, the use of a different sequence mixers inside of CAT can unlock the test-time control of efficiency with those sequence mixers_ (e.g., GDN in this case).
> >
> > We will update the submission with this new experiment.
> >
> > ### CAT can be a layer in any architecture
> > Finally, CAT can be _mixed-and-matched_ with existing efficient sequence mixers: the same principles of compression and decoding introduced in CAT can be used to instantiate CAT as a layer. This layer formulation of CAT can be used as a drop-in replacement layer in any architecture. Empirically we demonstrate CAT as a layer on the same MQAR task as the above experiment. The resulting model with CAT as layer solves the task.
> >
> > Hence, CAT as a layer can be used to build a new generation of hybrid architectures. One can even mix all three layers (dense attention, linear attention and CAT attention) in a single architecture to have benefits of all. **Further, CAT as a layer admits all the benefits of the CAT architecture including test-time control of efficiency, which most efficient layers do not provide.**
> >
> > We will update the submission with this new experiment.

---

> > > ### Author Response · Authors · 2025-11-25
> > >
> > > ### Summary
> > > Here, we summarize our general response. In short:
> > > - The **correct comparison between models is quality at given inference costs**, and raw parameters matter only so much as they affect inference costs
> > > - CAT achieves almost a **pareto frontier in quality-inference costs trade-off curve**
> > > - CAT provides **_test-time control_ of quality-inference costs trade-off** without any re-training, a capability absent in most efficient architectures
> > > - CAT is **complementary to existing sequence mixers**, and can be used as a drop-in replacement layer in any architecture

---

> > > > ### Author Response · Authors · 2025-11-25
> > > >
> > > > [1] Andy Jassy. “CEO Andy Jassy’s 2022 Letter to Shareholders.” Amazon / About Amazon, 2022.
> > > >  https://www.aboutamazon.com/news/company-news/amazon-ceo-andy-jassy-2022-letter-to-shareholders About Amazon
> > > >
> > > > [2] Monetizely. “Pricing AI Training vs Inference: Different Models for Different Phases.” Monetizely Blog, June 18, 2025.
> > > >  https://www.getmonetizely.com/articles/pricing-ai-training-vs-inference-different-models-for-different-phases Monetizely
> > > >
> > > > [3] Paul Bridi. “Managing Realtime AI Cost in Production: A Practical Guide.” Seldon – Take Control of ML and AI Complexity, 2025.
> > > >  https://www.seldon.io/managing-realtime-ai-cost-in-production-a-practical-guide/ Take Control of ML and AI Complexity+1
> > > >
> > > > [4] GMI Cloud. “Which Cloud Providers Maximize AI Inference ROI in 2025?” GMI Cloud Blog, 2025.
> > > >  https://www.gmicloud.ai/blog/which-cloud-providers-maximize-ai-inference-roi-in-2025 GMI Cloud
> > > >
> > > > [5] Amey Dhavle. “From Machine Learning to Small Language Models.” Medium, July 31, 2025.
> > > >  https://medium.com/@ameydhavle/from-machine-learning-to-small-language-models-7ce01227bdde Medium
> > > >
> > > > [6] Finbarr Taylor. “Large language models aren’t trained enough.” finbarr.ca Blog, February 27, 2023.
> > > >  https://finbarr.ca/llms-not-trained-enough/
> > > >
> > > > [7] [2402.18668] Simple linear attention language models balance the recall-throughput tradeoff
> > > >
> > > > [8] Yang, An, et al. "Qwen3 technical report." arXiv preprint arXiv:2505.09388 (2025).
> > > >
> > > > [9] Agarwal, Sandhini, et al. "gpt-oss-120b & gpt-oss-20b model card." arXiv preprint arXiv:2508.10925 (2025).
> > > >
> > > > [10] MoEs have the same inference costs as a dense transformer even if they have 10-20x more parameters than the dense transformer. This is possible because (i) model parameters occupy significantly less memory than the KV cache size used during generation (note that KV cache size is same in both MoEs and the corresponding dense transformer), and (ii) sparsity due to routing ensures computations remain unchanged.
> > > >
> > > > [11] Yang et al. 2024, Gated Delta Networks: Improving Mamba2 with Delta Rule
> > > >
> > > > [12] Gholami et al. 2024, AI and Memory Wall

---

> > > > > ### Author Response · Authors · 2025-11-27
> > > > > **updated the submission**
> > > > >
> > > > > We have revised the submission as follows:
> > > > >
> > > > > - Added a discussion that quality at a fixed inference cost is the appropriate comparison criterion (Section 4.1).
> > > > > - Updated Figure 1.
> > > > > - Added model-configuration details and in-context recall (Table 2) and language-modeling results (Tables 3 and 5) for the new pre-trained models (description of new models at: Section 4.2, Appendix D.1).
> > > > > - Included preliminary results showing that CAT acts as a meta sequence mixer (Appendix A.8) and that CAT can be instantiated as a layer within other architectures (Appendix A.7).
> > > > > - Added a discussion explaining why Dense 2D is an unfair comparison to CAT (Appendix A.5).
> > > > > - Added across-chunk analysis results (Appendix A.11).
> > > > >
> > > > > We hope these additions and clarifications address the reviewers’ concerns

---

### Author Response · Authors · 2025-11-25
**General Response 2: Parameter matched dense transformer is an unfair comparison to efficient baselines**

Some reviewers raise the concern that CAT is a weaker model in quality, especially when compared to a “more balanced”, parameter-matched dense transformer (Dense 2D).

We would first like to reiterate the point from our general response 1 ([here](https://openreview.net/forum?id=6rYa2BUnTt&noteId=3r23OwI7TD)): **raw parameter counts only matter insofar as they affect inference costs. The relevant comparison is what quality a model achieves under a fixed inference cost budget, and not whether two models have the same number of parameters.**

From this inference-cost perspective, Dense 2D is in fact an unfair comparison to CAT (or any other efficient baseline for that matter): it consumes ~4× more memory and requires ~3× more time to generate than even the least efficient CAT setting (CAT-4). A fair comparison would instead scale down a dense transformer until it is memory-matched to any CAT, say CAT-4 (e.g., by reducing hidden size or depth). Such a dense model would then run under the same memory budget as CAT-4 but would only be worse on real-world recall tasks and language modeling.

Indeed, when we compare our standard dense baseline (Dense D) with CAT, we find that CAT uses ~2-9× less memory, generates ~1.5-3× faster, and still matches Dense D on language-modeling evaluations. **_This supports our claim that CAT can match dense transformers in language modeling while being more efficient._**

Finally, we emphasize that _real-world practice already prioritizes inference cost over parameter counts._ Mixture-of-Experts (MoE) models are now widely deployed everywhere despite having 10–20× more parameters. When compared at equal total parameter counts, MoEs underperform their dense counterparts (Table 2 in [1]). By a purely “parameter-matching” logic, one can conclude that MoEs are “parameter inefficient” and thus not useful as architectures at all -- yet they are widely used in production _precisely because they deliver better quality at similar or lower inference cost._

CAT should be viewed in this same spirit: **_what matters is that, for a lower or fixed inference cost budget, CAT achieves competitive or better performance than dense transformers._**

In summary, Dense-2D is significantly more expensive to run than CAT and is not a fair comparison to CAT or to any other efficient model.
**We reported Dense 2D performance for transparency and completion of results, rather than as a direct comparison.** Our claims are grounded in comparisons at fixed inference costs, where CAT matches or improves on dense transformer performance. As we highlighted in our general response earlier, raw parameter counts are not the primary axis along which efficiency of a model should be judged.

We will add a discussion about this in the final manuscript to make this clear.

[1] Dai et al. 2024, DeepSeekMoE: Towards Ultimate Expert Specialization in Mixture-of-Experts Language Models

---

### Author Response · Authors · 2025-12-04
**Final response for AC**

As we approach the end of rebuttal, we provide a summary of the review process here for the AC.

# Positive aspects highlighted in the reviews
- **CAT provides a _knob_ to control efficiency:** All reviewers appreciated the **“concept of a single, adaptive model”** (Ue8h) that provides a **“clear knob” (Gs4J) to trade-off quality for efficiency directly at test-time without any re-training as “novel”** (Ue8h), **“useful and interesting”** (7wBq), making CAT **“practical for serving”** (KFwk) and deployment. They further added that this knob is **“absent in most existing efficient architectures”** (Ue8h, 7wBq), where “most efficient architectures choose the memory and compute budget a-priori” (7wBq).

- **CAT is a simple and scalable architecture:** Several reviewers appreciated the **“simple, effective and easy to grasp”** (Ue8h, McFn, KFwk) architecture of CAT, that offers **“distinctive advantages over prior approaches”** (McFn). The reviewers also appreciate the **“novel” (McFn) and “scalable training trick that avoids recurrence”** (Gs4J), allowing CAT to “maintain parallelism” (McFn) and scale to large datasets and model sizes, similar to a dense transformer. Further, they note that CAT requires “no handcrafted recurrent updates” (KFwk) due to **end-to-end learning.**

- **Exhaustive evaluations, ablations and implementation details**: The reviewers appreciate the **“comprehensive” (McFn) and “broad” (KFwk) evaluations on language modeling, synthetic and real-world recall, and long-context understanding.** They further appreciate the “ablations and design transparency” (Gs4J) that the paper provides, along with the extensive “implementation details (attention mask, KV reuse)” (KFwk).

---

> ### Author Response · Authors · 2025-12-04
>
> # Brief summary of the rebuttal
>
> We first provide a summary of the **main changes, clarifications** and **results** that we provided during the rebuttal.
>
> - We address the **main concern** about what the "right" comparison between models should be, all having different model architectures, model configurations and parameter counts.
> In our general response 1 ([here](https://openreview.net/forum?id=6rYa2BUnTt&noteId=3r23OwI7TD)), we argue why the raw parameter count is **not** the appropriate matching criterion between models.
> In short, inference costs of models dominate in the long run, and what matters for model deployment **is the quality at a desired inference cost, and not the raw parameter count itself.** The raw parameter count matters only so much as it affects the inference costs (both computations required and memory usage). **As a result, we compare different models by what quality (in-context recall) they obtain given an inference cost budget (specifically memory usage).** Please refer to general response 1 ([here](https://openreview.net/forum?id=6rYa2BUnTt&noteId=3r23OwI7TD)) for a detailed discussion.
>
> - We addressed concerns about “incomplete” baselines by **pre-training 7 additional models**: for each baseline, we change model configuration such as changing hybrid ratios or model dimension/recurrent state size etc. **as suggested by the reviewers**. **This results in a total of 12 models having different model types, configurations, parameter counts ranging from 300M to 820M (similar parameters as CAT) and _varying_ inference costs. Each baseline is ensured to have atleast two trained models with different model configurations for a broad and fair comparison. We compare all these 12 models, having varying inference costs, with a single CAT model.** Despite these additional models in comparison, **CAT still matches or outperforms all 12 models** across inference costs, using just a **single model only.** (Figure 1 **[here](https://ibb.co/d49gBVTH)** and in the paper shows the pareto-frontier of CAT in quality-inference costs trade-off curve.) Any prior approach requires pre-training from scratch to target a particular inference cost budget, whereas **CAT enables one to "traverse" this trade-off directly at test-time, without any training.** Please refer to general response 1 ([here](https://openreview.net/forum?id=6rYa2BUnTt&noteId=3r23OwI7TD)) for details regarding the newly pre-trained models.
>
> - We show CAT at its core is a **"meta-sequence mixer": meaning it is complementary to existing efficient architectures or sequence mixers, and can unlock controllable efficiency in any architecture or sequence mixers**. Our general response 1 ([here](https://openreview.net/forum?id=6rYa2BUnTt&noteId=3r23OwI7TD)) provides a preliminary result where CAT utilizes a GDN-Hybrid model as a decoder in CAT. Further, one can instantiate CAT as a layer and it can be used as a drop-in replacement layer in any architecture, meaning it can be **mixed-and-matched** with existing sequence mixers to create novel hybrid architectures. Our general response 1 ([here](https://openreview.net/forum?id=6rYa2BUnTt&noteId=3r23OwI7TD)) provides this result.
>
>
> We believe our rebuttal and the final version of the paper addresses all concerns raised by the reviewers, **clarifies any misunderstanding**, and strengthens the empirical support for our claims. We thank the reviewers for their thoughtful feedback.
>
> We believe this paper contributes a **new and simple architecture with a property absent in most efficient architectures**, as highlighted by all reviewers. This property of controllable efficiency makes **CAT a very attractive architecture to be deployed at scale in the real world and production scenarios.**
>
> We have updated the paper and hope the **AC will consider our responses and our clarifications to the reviewers’ misunderstandings when making the final decision.**
>
> We now present how we addressed the main concerns in the rebuttal, followed by a summary of how we addressed reviewer specific concerns.

---

> > ### Author Response · Authors · 2025-12-04
> >
> > # Main concerns raised in the initial round, and how we addressed them
> >
> > >>> 1. **Unfair comparisons** and/or **provide apples-to-apples comparison** (raised by all reviewers: Ue8h, 7wBq, KFwk, McFn, Gs4J)
> >
> > All reviewers raised the concern that CAT models were not “parameter matched” to the baselines and suggested that the differences in parameter counts makes the comparison unfair.
> >
> > In our general response 1 ([here](https://openreview.net/forum?id=6rYa2BUnTt&noteId=3r23OwI7TD)), we discuss why the raw parameter count is not the appropriate matching criterion to compare different models. In short, when language models are deployed, **inference costs dominate** (both compute and memory), and **raw parameter counts only matter so much as their effect on inference cost.** A good example of this is widely deployed Mixture-Of-Experts (MoEs), where 10-20x more parameters does not mean increased inference costs. As a result, the right comparison among different models (all having different model types, configuration and parameter counts) is what quality one obtains at a given inference cost.
> > Hence, we compare different models by **matching inference costs.**
> >
> > Further, to enable a broader and fair evaluation, we pre-trained new baselines (7 additional models) by changing their model configuration (e.g. by tweaking hybrid ratios or **increasing model/recurrent state dimension** as **suggested by reviewer McFn**)
> >
> > **This increased our baselines from 5 models in our initial draft, to a total of 12 models all having different model architectures and types, model configurations, parameter counts ranging from 300M to 820M (similar parameters as CAT), and varying inference costs.**
> >
> > **Figure 1 ([here](https://ibb.co/d49gBVTH))** shows that _despite_ these additional models in comparison, CAT is still **similar or better in quality than every model across varying inference costs**, achieving a pareto-frontier in quality-inference costs trade-off curve. **More importantly, CAT achieves this pareto-frontier using a single model only.**
> >
> > Finally, to drive home the point of comparison on inference costs, we raised the following question to the reviewers in our general response 1, and were curious about their answers on it during the rebuttal:
> > **Suppose model A has more parameters than model B, then if model A outperforms model B using lower inference costs, does it matter that model A has more parameters than model B? Which model should one deploy: model A or B?**
> >
> > > As an aside, note that CAT focuses on reducing inference costs through efficient sequence mixing, whereas MoEs focus on adding more parameters in the feed forward layer without increasing inference costs. As a result, MoEs are complementary and can be applied to any sequence mixer, including the ones used inside of CAT. We have added this point in our related work.

---

> > > ### Author Response · Authors · 2025-12-04
> > >
> > > >>> 2. **CAT is inferior to parameter matched dense transformer (Dense 2D)** (raised by reviewers Ue8h, McFn)
> > >
> > > Several reviewers (Ue8h, McFn) were concerned that CAT is inferior in quality compared to a parameter matched dense transformer (Dense 2D), and that this inferior quality “contradicts the main claims from the abstract” (Ue8h).
> > >
> > > In short, our general response 2 ([here](https://openreview.net/forum?id=6rYa2BUnTt&noteId=WgvmGO1piP)) addresses this concern where we state that a parameter matched dense transformer (Dense 2D) is an unfair comparison to CAT since it requires significantly more inference costs (4x more memory and 3x more time). We reported Dense 2D for **transparency and completeness of the results, and NOT as a direct comparison with CAT.**
> > >
> > > Our comparisons are grounded in _what quality a model obtains at a given inference cost._ To enable a fair comparison, one would scale down a dense transformer such that it requires similar inference costs to any CAT (or scale up CAT such that it takes similar inference cost as Dense 2D). Indeed, when we compare Dense D/2 to an inference costs matched CAT (i.e. CAT-4), CAT outperforms Dense D/2. Interestingly, CAT-4 also outperforms Dense D on real-world in-context recall tasks while being 2x memory efficient and 1.5x faster.
> > >
> > > Further, CAT(-4/8/16/32) matches Dense D in language modeling while requiring significantly lower inference costs (2-9x memory efficient and 1.5-3x faster). This supports the claim in our abstract.
> > >
> > > A good example where parameter matching may lead to incorrect conclusions is again MoEs. When compared at equal total parameter counts, MoEs underperform their dense counterparts (Table 2 in [1]). By a purely “parameter-matching” (Ue8h, McFn) logic, one may conclude that MoEs are “parameter inefficient” (Ue8h) compared to a dense transformer, and thus not useful as architectures at all. Yet they are widely used in production precisely because they deliver better quality at similar or lower inference cost.
> > >
> > > **CAT should be viewed in this same spirit: CAT achieves competitive or better performance than dense transformers while using lower inference costs.**
> > >
> > > We have added a discussion in Appendix A.5 about this.
> > >
> > > [1] Dai et al. 2024, DeepSeekMoE: Towards Ultimate Expert Specialization in Mixture-of-Experts Language Models
> > >
> > >
> > > >>> 3. **CAT’s practical efficiency benefits might be moderate at best** and **layerwise schemes** may offer better quality-inference cost trade-offs (reviewer McFn)
> > >
> > > Reviewer McFn raised the concern that the “layerwise schemes” of hybrid architectures may offer quality-inference costs trade-offs. To address this, we pre-trained 7 additional models including new hybrid architecture baselines by tweaking their hybrid ratios, model dimensions and sliding window sizes as **suggested by the reviewer McFn themselves.**
> > >
> > > **We find that CAT still matches or outperforms the hybrid models** across different “layerwise schemes” having varying inference costs, using a single model only.
> > >
> > > Further, note that any prior baseline including hybrids require pre-training from scratch and careful tuning of hybrid ratios (e.g. setting it to 1:4, 1:7, or any 1:X etc.) to target any different inference cost budget. **CAT avoids this tuning all together and provides this capability to change inference costs directly at test-time with no training.**
> > >
> > > Importantly, we would like to highlight that CAT is a “meta sequence mixer” and is actually complementary to any existing or future efficient sequence mixers or architectures. CAT has two components: a compressor and a decoder – each of these could make use of any sequence mixers, such as linear attention (e.g. GDN or GatedDeltaNet): meaning rather than CAT being a strict competitor to GDN (or any other efficient sequence mixer), CAT is complementary to existing sequence mixers.
> > > We empirically show this result in our general response 1. Further, the use of a different sequence mixers inside of CAT can unlock the test-time control of efficiency with those sequence mixers (e.g., GDN in this case).
> > >
> > > Finally, CAT can be **mixed-and-matched with existing efficient sequence mixers**: the same principles of compression and decoding introduced in CAT can be used to instantiate CAT as a layer. This layer formulation of CAT can be used as a drop-in replacement layer in any architecture. **CAT as a layer admits all the benefits of the CAT architecture including test-time control of efficiency, which most efficient layers do not provide.  We show this result in our general response 1 ([here](https://openreview.net/forum?id=6rYa2BUnTt&noteId=3r23OwI7TD)) and provide more details there.**

---

> > > > ### Author Response · Authors · 2025-12-04
> > > >
> > > > # Reviewer specific concerns
> > > > We provide a summary of **each** of the reviewers specific concerns, and how we addressed it:
> > > >
> > > > ## Reviewer Gs4J
> > > > >> Concerns about an “apples‑to‑apples comparison” between different models due to parameter differences
> > > >
> > > > Addressed in general response 1 ([here](https://openreview.net/forum?id=6rYa2BUnTt&noteId=3r23OwI7TD)), where we clarify that the correct “apples‑to‑apples comparison” between models is at matched inference costs.
> > > >
> > > > >> Concerns about “parameter‑matched dense (2D)” outperforming CAT
> > > >
> > > > Addressed in general response 2, where we discuss Dense 2D is an unfair comparison to CAT since Dense 2D has significantly more inference costs. A fair comparison would involve scaling down dense transformer (or scaling up CAT) that use similar inference costs.
> > > >
> > > >
> > > > >> The reviewer asked for an analysis on how the perplexity changes across chunk boundaries.
> > > >
> > > > We added analysis for perplexity changes across chunk boundary in the reviewer response and Appendix A.11.
> > > >
> > > > >> Performance of CAT beyond 4K context length
> > > >
> > > > We provide scaled up results on **8K context lengths on a larger pre-training budget of 30B tokens and larger models** (increased layers to 18). **CAT still outperforms existing baselines on 8K context lengths.**
> > > >
> > > > ## Reviewer McFn
> > > >
> > > > >> Concerned about “Unfair comparison” due to parameter differences, and adds that this makes the baselines “meaningless” and “incomplete and weak”. The baselines could be “similarly scaled up in parameters” while still retaining efficient inference costs.
> > > >
> > > > We address the concern about “Unfair comparison” in general response 1 ([here](https://openreview.net/forum?id=6rYa2BUnTt&noteId=3r23OwI7TD)), where we argue why the raw parameter count is not the appropriate matching criterion to compare different models, and that the right comparison is what quality a model obtains at a given inference cost.
> > > >
> > > > We further address the concern about “meaningless” baselines where we **“similarly scale up parameters” resulting in a total of 12 models, spanning different model types (linear, dense attention and hybrids), configurations (different hybrid ratios, recurrent state size, model dimension), parameters (ranging from 300M to 820M), that enable a fair and broader evaluation with CAT.**
> > > >
> > > > We add models with configurations **suggested by the reviewer** to address “Weak and incomplete baselines”, specifically we pre-trained GDN-H G 1:7, GDN-H G 1:7 2D, GDN-2x.
> > > >
> > > > **Despite the additional models in comparison including those that have similar parameters to CAT, CAT still matches or outperforms these models across varying inference costs, using a single model only.**
> > > >
> > > > >> Concerned that “CAT shows some disadvantage” in quality when compared to a “more balanced comparison” with a parameter matched dense transformer (Dense 2D)
> > > >
> > > > We address this in global response 2, where we discuss Dense 2D is an unfair comparison to CAT since Dense 2D has significantly more inference costs (4x more memory, 3x more generation time). A fair comparison would involve scaling down dense transformer (e.g. Dense D/2), where CAT-4 outperforms Dense D/2 while using the same inference costs.
> > > >
> > > > We further clarify that such “layerwise scheme” of hybrid architectures that can extend efficiency of hybrids are **complementary to CAT since CAT’s decoder can use GDN-Hybrid architecture (general response 1)**, helping extend “practical efficiency benefits” of hybrid architectures to CAT. As a result, CAT can be thought of as a **“meta sequence mixer”, where instead of being a competitor, CAT is complementary.**
> > > >
> > > >
> > > > ## Reviewer Ue8h
> > > > >> Concerns about “invalid comparison in the main results” due to parameter differences between CAT and the baselines.
> > > >
> > > > We provide a discussion in our general response 1 ([here](https://openreview.net/forum?id=6rYa2BUnTt&noteId=3r23OwI7TD)), where we state that parameter count is not the right comparison criteria: what matters is what quality a model obtains given an inference cost. Raw parameter counts only matter so much as their effect on inference costs.
> > > >
> > > > >> Concerned that since parameter matched dense transformer (Dense 2D) outperforms CAT, this observation “contradicts the main claims from the abstract”.
> > > >
> > > > We provide a discussion in general response 2, where we state that Dense 2D is an unfair comparison to CAT. The correct comparison is what quality a model obtains at a given inference cost. Dense 2D has significantly more inference costs (4x more memory, 3x more generation time).
> > > >
> > > > When CAT is compared to an inference cost matched dense transformer (Dense D/2), CAT-4 outperforms Dense D/2. Interestingly, CAT-4 even outperforms Dense D. Further, CAT matches Dense D on language modeling while being faster and memory efficient (having lower inference costs), supporting the claims in our abstract.

---

> > > > > ### Author Response · Authors · 2025-12-04
> > > > >
> > > > > ## Reviewer 7wBq
> > > > >
> > > > > >> Concerns about finding it “difficult to isolate the contribution of the architecture” and an “apples-to-apples” comparison “would strengthen the case”
> > > > >
> > > > > We clarify comparison across different models is matching inference cost. This matching helps “isolate the contribution of the architecture” where every model has different model types, configurations and parameters.
> > > > >
> > > > > >> Analysis on how the perplexity changes across chunk boundaries.
> > > > >
> > > > > We added analysis for perplexity changes across chunk boundary in reviewer response and Appendix A.11.
> > > > >
> > > > > >> Error analysis for Dense and GDN-Hybrid on FDA and NIAH-N tasks in the rebuttal.
> > > > >
> > > > > We provided generations for both models, where we observed some interesting trends: these models either get the first digit or part of the answer correct, and hallucinate the rest, or they retrieve text close to the answer text, but not the answer exactly.
> > > > >
> > > > > >> “Surprised” that CAT works well on the MQAR task where even “compressing 4 tokens could drop a lot of precision”
> > > > >
> > > > > We provide an evaluation by “scaling up keys and values” in Figure 4 in our submission to upto 1K sequence length (4x the standard length in this task), showing empirically that **CAT does learn to decode values accurately despite compression, even outperforming GDN and Mamba2 at longer contexts while using the same memory.**
> > > > >
> > > > >
> > > > > ## Reviewer KFwk
> > > > > >> Concerns about the “Mismatched capacity (parameters) in core tables” and that this parameter difference clouds the “architecture vs scale” effects.
> > > > >
> > > > > We address this in general response 1 ([here](https://openreview.net/forum?id=6rYa2BUnTt&noteId=3r23OwI7TD)), where we state that the correct comparison is what quality a model obtains at given inference cost.
> > > > >
> > > > > Further, matching inference cost across different models helps address the “architecture vs scale” question, where **even “scaled(-up) baselines” are considered valid comparisons as long as they are inference cost matched with CAT.**
> > > > >
> > > > > >> Detailed accounting for training cost breakups, and component wise time accounting during generation
> > > > >
> > > > > We provide these in our rebuttal in the reviewer response, and will add these breakups in our final submission.
> > > > >
> > > > > >> The reviewer was also interested in an ablation where learned compression is replaced with pooling that we provide in our rebuttal
> > > > >
> > > > > We perform this experiment and find that this CAT model performs worse. On WikiText dataset, perplexity is 21.47 vs 17.7 in the case where compression is learnt using a transformer.

---

> > > > > > ### Author Response · Authors · 2025-12-04
> > > > > >
> > > > > > ## Updated the paper
> > > > > >
> > > > > > We have updated the paper with the said changes. We believe this address all reviewers' concerns and misunderstanding.

---

### Meta-Review · Area_Chair_VhEh · 2025-12-22

**Summary:**

The paper introduces a new technique they name CAT (Compress and attend Transformers) and that enables test-time control of the quality/inference-cost trade-off from a single trained model by attending to compressed representations of past context.

The reviews agree that CAT introduces a novel and practical idea: a knob that can be chosen at test-time to trade off quality for efficiency without retraining. I think this point is convincing.

However, a key concern raised by **all** reviewers is the fairness of the comparisons presented in the paper: reviewers criticize the use of larger CAT models against smaller baselines and argue that parameter-matched dense transformers outperform CAT on standard language modeling tasks.
In response, the authors argue that inference cost (not parameter count) is the correct comparison axis. I do think the position of the authors is defensible and perhaps well-aligned with deployment reality, but I think both axes should have been reported more cleanly in the paper. I find it surprising that the authors consistently rejected parameter matching as the correct criterion, I think this is subject to discussion. My view is that the best option would be to provide the experiments requested by the authors and discuss this topic in an open manner in the paper.

I have to admit that this paper is difficult to judge, especially given the limited interaction during the review phase and the reassignment to new ACs. Overall, I believe the proposed method is promising and addresses a practically important problem with an original idea. At the same time, I agree with the reviewers that the paper would benefit from more transparency and nuance in how comparisons to the baselines are presented and interpreted. Given the strong divergence of opinions and the remaining concerns around evaluation clarity, I have decided not to recommend acceptance at this time. That said, I sincerely encourage the authors to address this feedback, as I believe a revised version that more clearly positions CAT within both parameter- and inference-cost-based comparisons could make a strong contribution in a future submission.

**Reviewer Concerns:**

Most technical concerns were addressed except for the key concern regarding the comparison to smaller baselines. The authors consistently rejected parameter matching as the correct criterion, instead of trying to engage in a more open discussion. This is somewhat concerning to me which led me to the decision of rejecting the paper.

**Reviewer Scores:**

I do not expect the reviewers to fully agree with the authors’ main argument regarding the central issue raised above. While some secondary concerns may have been addressed, they appear less important than this primary point of disagreement.

---

### Decision · Program_Chairs · 2026-01-26

Reject